# Aggregation promotes charge separation in fullerene-indacenodithiophene dyad

Chong Wang [1,2], Bo Wu [1,2] ✉, Yang Li[3], Shen Zhou [4], Conghui Wu[5], Tianyang Dong[1], Ying Jiang[1,2], Zihui Hua[1,2], Yupeng Song[1,6], Wei Wen [1,7], Jianxin Tian[1,2], Yongqiang Chai[8], Rui Wen [1,2] & Chunru Wang [1,2] ✉

Fast photoinduced charge separation (CS) and long-lived charge-separated state (CSS) in small-molecules facilitate light-energy conversion, while simultaneous attainment of both remains challenging. Here we accomplish this through aggregation based on fullerene-indacenodithiophene dyads. Transient absorption spectroscopy reveals that, compared to solution, the CS time in aggregates is accelerated from 41.5 ps to 0.4 ps, and the CSS lifetime is prolonged from 311.4 ps to 40 μs, indicating that aggregation concomitantly promotes fast CS and long-lived CSS. Fast CS arises from the hot charge-transfer states dissociation, opening up additional resonant channels to free carriers (FCs); subsequently, charge recombination into intramolecular triplet CSS becomes favorable mediated by spin-uncorrelated FCs. Different from fullerene/indacenodithiophene blends, the unique CS mechanism in dyad aggregates reduces the long-lived CSS dependence on molecular order, resulting in a CSS lifetime 200 times longer than blends. This endows the dyad aggregates to exhibit both photoelectronic switch properties and superior photocatalytic capabilities.

Photoinduced electron transfer is pivotal for light-energy conversion, which is ubiquitous in photovoltaic conversion and photosynthetic and photocatalytic reactions[1–3]. To attain efficient energy conversion, the realization of fast charge separation (CS) and long-lived charge-separated states (CSS) is highly desired, which facilitate more opportunities for excited electrons and holes to engage in subsequent photoreactions[1,4,5]. Serving as fundamental building blocks in photoelectronic materials, organic small molecules (SM) covalently linked by donor-acceptor (D-A) structures have well-defined structures, easy purification, and simplified manufacturing technology. Moreover, they guarantee straightforward molecular design and unambiguous photoelectric conversion mechanisms[6,7]. However, achieving fast CS and long-lived CSS concomitantly in SMs presents a challenge, because charge separation and recombination commonly occur in the Marcus normal and inverted regions, respectively[1]. Although there have been explorations in molecular design targeting these characteristics, the inherent challenges stemming from complex molecular design and energy loss during multistep CS processes currently impede further application[1,8,9]. Developing simple and efficient charge separation regulation

[1]Beijing National Laboratory for Molecular Sciences, Key Laboratory of Molecular Nanostructure and Nanotechnology, Institute of Chemistry, Chinese Academy of Sciences, Beijing 100190, China. [2]University of Chinese Academy of Sciences, Beijing 100049, China. [3]School of Science, Beijing University of Posts and Telecommunications (BUPT), Beijing 100876, China. [4]College of Science, Hunan Key Laboratory of Mechanism and Technology of Quantum Information, National University of Defense Technology, Changsha 410003, China. [5]Spin-X Institute, School of Chemistry and Chemical Engineering, South China University of Technology, Guangzhou 511442, China. [6]Key Laboratory of Photochemical Conversion and Optoelectronic Materials and CityU-CAS Joint Laboratory of Functional Materials and Devices, Technical Institute of Physics and Chemistry, Chinese Academy of Sciences, Beijing 100190, China. [7]Beijing National Laboratory for Molecular Sciences, CAS Key Laboratory of Organic Solids, Institute of Chemistry, Chinese Academy of Sciences, Beijing 100190, China. [8]Department of Chemistry and Pharmacy and Interdisciplinary Center for Molecular Materials (ICMM), Friedrich-Alexander-Universität Erlangen-Nürnberg, Erlangen 91058, Germany. ✉e-mail: zkywubo@iccas.ac.cn; crwang@iccas.ac.cn

strategies, coupled with an in-depth exploration of the fundamental mechanisms, remains worthy of further investigation[10,11].

Typical photophysical processes for photoelectronic materials are shown in Fig. 1a[12,13]. CS and charge recombination (CR) processes are the focus of our study. Generally, because of the forbidden transition, the triplet charge-separated state ($^3$CSS) has a longer lifetime than the singlet CSS ($^1$CSS)[12], but its generation usually requires spin-flip. As a result, intramolecular $^3$CSS generation is disadvantageous in terms of thermodynamic and kinetic competition, resulting in limited yield, and most electrons and holes recombine into ground-state (GS) from the short-lived $^1$CSS instead[14]. Overcoming the spin limitations promotes the $^3$CSS generation, thus slowing down CR. This inspired us to investigate photogenerated free carriers (FCs), which are spin-uncorrelated and commonly exist in aggregates[12,15,16]. Compared to individual molecules, aggregates such as thin-films, nanoparticles, and heterojunctions exhibit distinct photophysical properties owing to enhanced intermolecular interactions[17,18], including charge transfer, separation, and recombination processes. In this scenario, FCs generation can be accelerated by hot charge-transfer state (CTS) dissociation owing to the promoted resonance with CSS (Fig. 1a)[15,16,19,20]. Furthermore, the spin-uncorrelated FCs can recombine into $^3$CSS with a considerable yield mandated by spin statistics, thus prospectively behaving as long-lived CSS[12,21,22]. Therefore, fast CS and long-lived CSS are expected to be achieved in aggregates.

In this study, we designed two covalent fullerene ($C_{60}$)-indace-nodithiophene (IT) SMs (Fig. 1b, IT-$C_{60}$), named ID and IB, and studied the photoinduced charge separation processes in solution, poly(methylmethacrylate) (PMMA)-diluted films and aggregated films. Our design rationale is based on the following considerations. (I) Fullerene, represented by $C_{60}$, is a prominent electron acceptor due to its high electronic affinity, low reorganization energy, and three-dimensional electron coupling properties, making it a promising candidate for the CS mechanism study[1,23–25]. (II) IDTT and IBDT are rigid fused-ring donors with strong optical absorption and efficient electron transport properties (Fig. 1b)[26,27]. Furthermore, the similar structures with diverse fused-ring lengths allow for generalizing the CS regularity.

Employing transient absorption and pulsed electron paramagnetic resonance, we discovered that aggregation concomitantly promotes fast CS and long-lived $^3$CSS in aggregates, benefitting from the hot CTS dissociation and free carrier mediation, respectively. The long-lived intramolecular $^3$CSS of dyads is mediated by FCs recombination, which differs from directly blended systems of donor/acceptor. This reduces the dependence of long-lived CSS on molecular order in dyads, making it easier to obtain long-lived CSS than blends. Accordingly, it endows dyads with the photoelectric switch features. High-energy photons excitation in aggregates can open pathway from intermolecular to intramolecular CSS, thereby obtaining both fast CS and long-lived CSS; while this pathway is closed using low-energy photons excitation or in non-aggregated systems. Additionally, the CSS lifetime of dyad aggregates is 200 times longer than that of blends, enabling dyads to store more redox equivalents confirmed by the photocurrent measurements, thus possessing superior photocatalytic capabilities. This work reveals the pronounced role of aggregation in promoting CS in organic SM dyads, and provides guidance for the application of SM-based photoelectronic materials.

## Results
### Synthesis and characterization

Two covalent IT-$C_{60}$ SMs, ID and IB (Fig. 1b), were synthesized according to the 1,3-dipolar cycloaddition reaction (prato reaction). The detailed synthesis, purification, and characterization steps are listed in Supplementary Notes 1–4 (Supplementary Figs. 1–5). Herein, o-dichlorobenzene (o-DCB) was selected as the solvent to study the non-aggregated system. The aggregated thin films were prepared by spin-coating a quartz substrate (Fig. 1c). Atomic force microscopy (AFM) and transmission electron microscopy (TEM) were used to understand the morphology and microstructure of the film, which revealed a smooth and uniform surface, indicating the homogeneity of the aggregates (Supplementary Fig. 6). Additionally, we fabricated the PMMA-doped films and the D/A directly blended films to compare their photophysical properties differences.

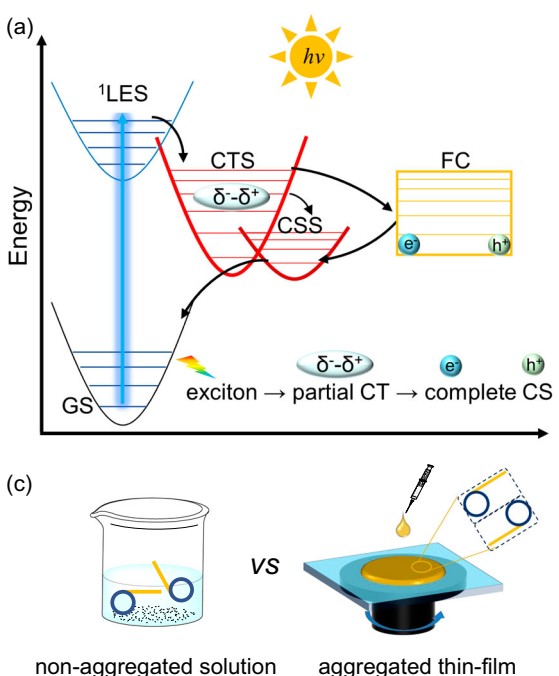

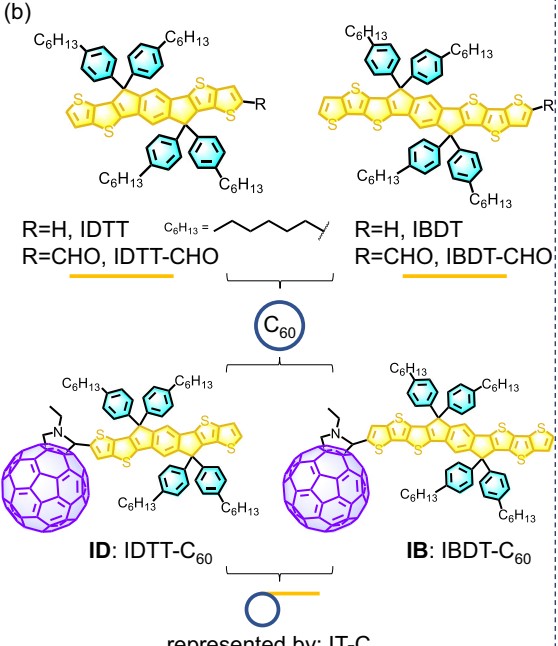

**Fig. 1 | Photophysical processes and molecular structures. a** Schematic of typical photophysical processes in photoelectronic materials. Here the abbreviations have the following meanings. GS: ground state; LES: localized excited state; CT: charge transfer; CS: charge separation; CTS: charge-transfer state; CSS: charge-separated state; FCs: free carriers. **b** Molecular structures of ID, IB and the precursors studied in this work. **c** Schematic of solution and thin-film.

## Study of steady-state spectra and electrochemistry

Steady-state absorption spectra were measured in *o*-DCB and film. The maximum absorption energy of IB was lower than that of ID, benefiting from the expansion of the conjugated fused-ring skeleton, which broadened the absorption range (Fig. 2a and Supplementary Figs. 7 and 8). The concentration dependence ($2 \times 10^{-5}$–$2 \times 10^{-4}$ M) of the absorption spectra confirmed the absence of intermolecular interactions in solution (Supplementary Fig. 9). When PMMA is doped to dilute the film, the absorption spectra closely resemble those in solution, indicating that intermolecular interactions are suppressed after PMMA dilution, exhibiting individual molecular features. In contrast, the pure films exhibited tailing and broadened absorption band in the visible region, indicating that aggregation enhances intermolecular coupling and creates opportunities for intermolecular CS.

Identifying radical ion absorption is a prerequisite for determining charge separation. We monitored the absorption spectral changes upon chemical oxidation by tris(4-bromophenyl)-ammoniumyl hexachloroantimonate ("Magic Blue," MB) in *o*-DCB[28]. Following the addition of MB, some discernible changes, 620 and 1250 nm for ID, and 680 and 1450 nm for IB, were observed. These can be assigned to the absorption after one-electron oxidation, namely radical cation (hole) absorption (Fig. 2b and Supplementary Fig. 7).

For a preliminary assessment of the excited-state behavior, photoluminescence (PL) measurements were conducted. In *o*-DCB, when the donor was selectively excited at 400 nm, apart from the donor's local emission around 500 nm, we also observed the emission of $C_{60}$ around 720 nm in both ID and IB (Fig. 2 and Supplementary Fig. 7). This indicates an energy-transfer pathway from the donor to $C_{60}$ existing in solution, which is thermodynamically feasible due to the lower singlet-state energy of $C_{60}$[23]. The PMMA-diluted films, as expected, show the

similar emission behaviors to solution, with distinct local emission for IT and $C_{60}$. Here the $C_{60}$ emission are more obvious than in solution, possibly due to the restriction of molecular motion, reducing energy loss during the donor-to-acceptor energy transfer. Interestingly, no local emission appeared in the film, but instead, a broad emission peak around 900 nm was observed in the NIR region, covering a range from 700 nm to 1200 nm. The appearance of NIR emission features after aggregation, along with the significant redshift compared to local emission, indicate that this emission is an intermolecular process, most likely due to intermolecular electron-hole recombination and similar with the blend films (Supplementary Fig. 10)[29,30]. The PL lifetime of the film is approximately 5 ns for both ID and IB, longer than the local emission lifetime in solution (~1 ns) (Supplementary Fig. 11).

Electrochemical analysis revealed that IBDT exhibited a stronger electron-donating capability than the IDTT donor, thus offering a larger driving force for CS in solution (Supplementary Note 7, Supplementary Figs. 12 and 13, Supplementary Tables 1 and 2). On the basis of the electrochemical analysis, the CSS energy levels in *o*-DCB and pure films are both lower than the singlet and triplet excited-state of $C_{60}$ ($^{3}C_{60}^{*}$). This suggests that charge separation is thermodynamically feasible, while charge recombination to the $^{3}C_{60}^{*}$ is thermodynamically unfavorable.

## Transient absorption investigation

To further investigate the photophysical processes, femtosecond transient absorption (fs-TA) was employed (Fig. 3). For ID, upon 410 nm excitation in *o*-DCB, the excited state absorption (ESA) around 730 nm generated within 0.1 ps, accompanied by the stimulated emission (SE) around 500 nm, can be attributed to the localized excited state (LES) of the precursor IDTT (Supplementary Fig. 14). From 1 ps to 50 ps, the gradually generated ESA at 620 nm and 1250 nm

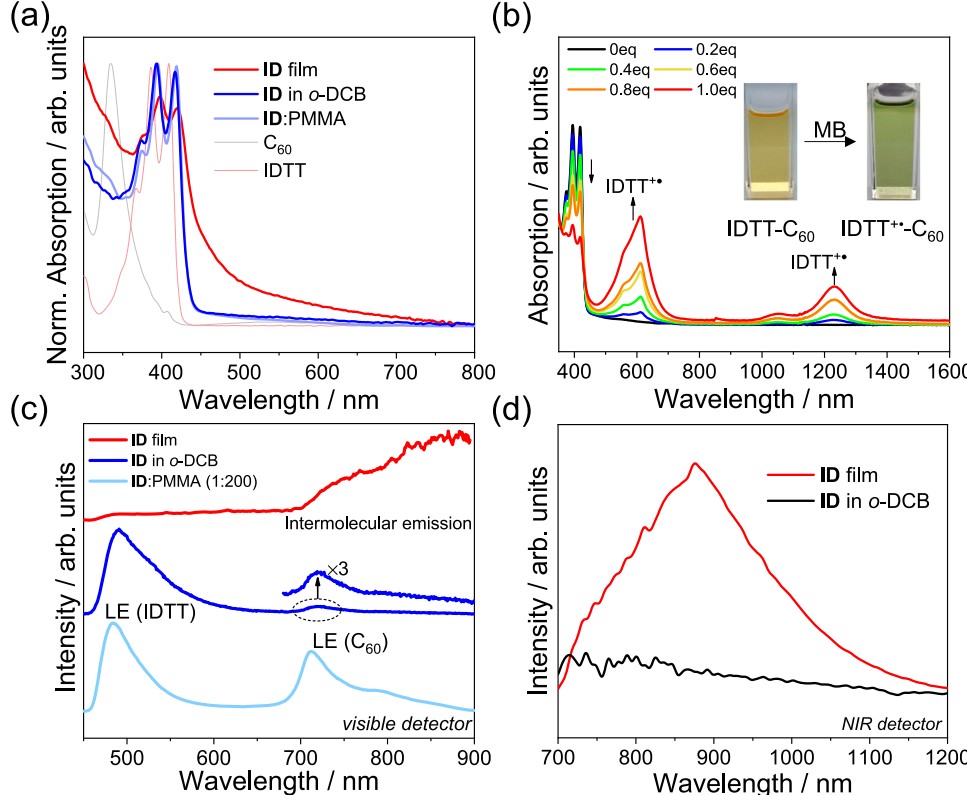

**Fig. 2 | Steady-state spectra. a** Normalized (Norm.) absorption spectra of ID in *o*-DCB and film, PMMA-diluted ID film (ID:PMMA = 1:200, w:w), and the precursors IDTT and $C_{60}$ in *o*-DCB. **b** Change in the spectra upon chemical oxidation under the addition of 0 to 1 equivalent (equiv) "Magic Blue" (MB) in *o*-DCB for ID; inset: photographs of the color change before (left) and after (right) oxidation. **c** Photoluminescence spectra of ID film, ID in *o*-DCB, and PMMA-diluted ID film, excited at 400 nm and recorded by visible detector. **d** Photoluminescence spectra of ID in *o*-DCB and film recorded by NIR detector.

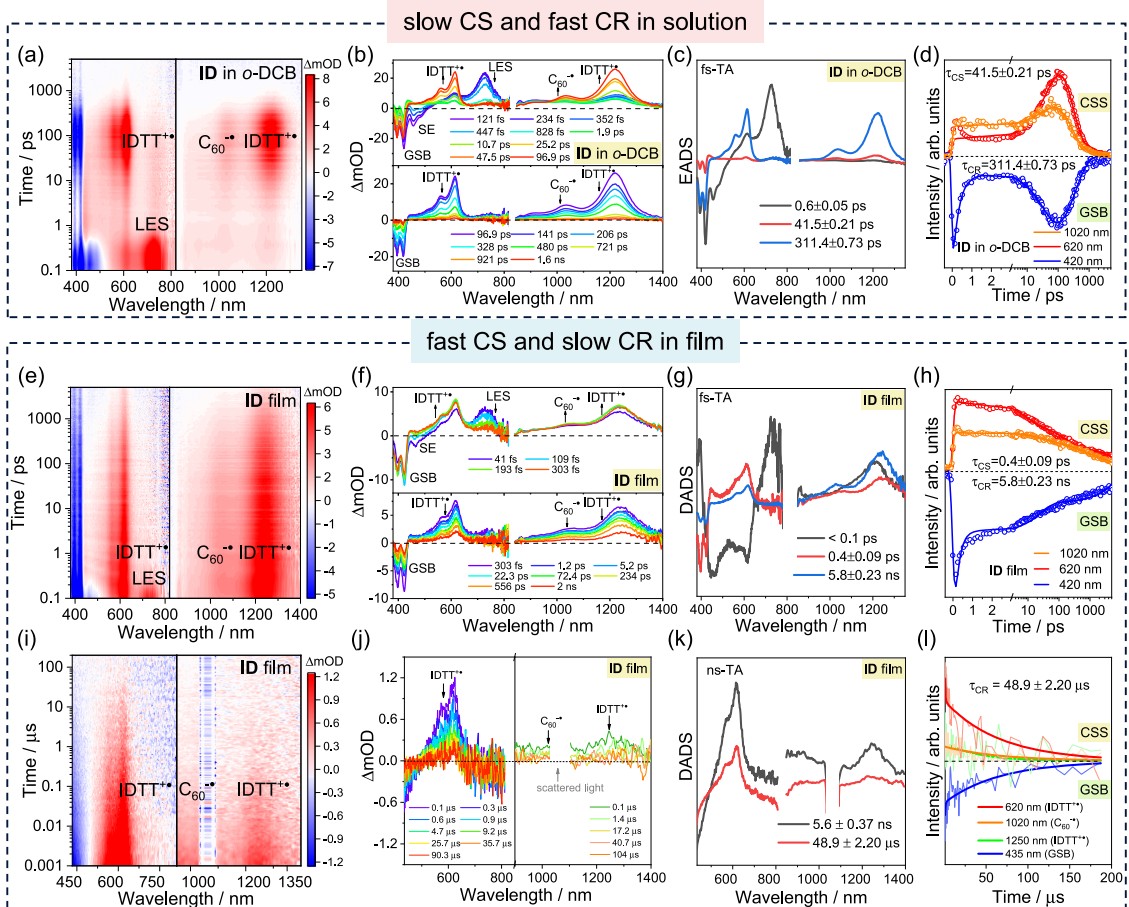

**Fig. 3 | Transient absorption (TA) of ID. a** Contour plot of femtosecond TA (fs-TA) excited at 410 nm in *o*-DCB. **b** Selected TA spectra (TAS) in *o*-DCB. **c** Evolution-associated difference spectra (EADS) of ID in *o*-DCB. **d** Kinetics of charge-separated state (CSS) and ground-state bleaching (GSB), and the fitted results of ID in *o*-DCB. **e** Contour plot obtained from fs-TA of ID film following 410 nm excitation. **f** Selected TAS of ID film. **g** Decay-associated difference spectra (DADS) of ID film.

**h** Kinetics of CSS and GSB, along with the fitted results of ID film obtained from fs-TA. **i** Contour plot obtained from nanosecond-TA (ns-TA) of ID film following 410 nm excitation. **j** Selected TAS of ID film at visible and NIR regions obtained from ns-TA. **k** DADS of ID film obtained from ns-TA. **l** Kinetics of CSS and GSB within 200 μs, along with the fitted results of ID film obtained from ns-TA.

reflects hole absorption (IDTT⁺˙, Fig. 2b). Coupled with the absorption of $C_{60}^{\cdot}$ (around 1020 nm, Supplementary Fig. 15), the typical absorption features of cation and anion, and the consistent kinetics, act as the evidence of the charge separation occurrence. The CSS reached its maximum absorption intensity within 100 ps and then decayed within 1 ns. Notably, within the initial 2 ps, charge separation did not occur according to the negligible hole absorption, whereas the LES were rapidly quenched, with a new weak ESA appearing at 440 nm (Fig. 3b). This reflects energy transfer (EnT) from the donor to $C_{60}$, which can be confirmed by selectively exciting $C_{60}$ at 350 nm (Supplementary Figs. 16 and 17). Moreover, the steady-state PL spectra in solution revealed the existence of an energy-transfer pathway as well (Fig. 2c). Fast energy transfer from the donor to the acceptor results in rapid quenching of the GSB and SE of the donor. Therefore, the evolution of the excited states in *o*-DCB involves (i) energy transfer from the donor to the acceptor, (ii) charge separation, and (iii) charge recombination. Employing global analysis based on the three-component model, the evolution-associated difference spectra (EADS) and corresponding time-constants were obtained as $\tau_{EnT} = 0.3$ ps, $\tau_{CS} = 41.5$ ps, and $\tau_{CR} = 311.4$ ps. We noted similar charge separation kinetics under excitation at 410 or 350 nm, suggesting that in solution, charge separation from the low-energy LES of $C_{60}$ acts as the rate-determining step (Supplementary Fig. 18).

The excited-state evolution processes of IB are generally analogous to those of ID, as discussed in Supplementary Note 8

(Supplementary Figs. 19–24). The times of charge separation (24.8 ps) and recombination (119.0 ps) in IB were shorter than those in ID, because of the stronger electron-donating ability of IBDT resulting in a larger driving force during CS and a smaller driving force during CR, which are located in the Marcus normal and inverted regions, respectively. We also selected benzonitrile (PhCN) (Supplementary Figs. 25 and 26) to provide larger driving force for CS. As expected, the CS and CR processes of both ID and IB accelerated, with $\tau_{CS} = 14.8$ ps and $\tau_{CR} = 81.0$ ps for ID, and $\tau_{CS} = 8.9$ ps and $\tau_{CR} = 22.5$ ps for IB.

While achieving the concomitant fast CS and slow CR are difficult in solution, there are impressive results in film (Fig. 3e–l). Upon 410 nm excitation, the LES of donor resolved within 0.1 ps. In contrast to solution, the deactivation of LES in the film was faster, and the cation (IDTT⁺˙) and anion ($C_{60}^{\cdot}$) absorption reached their maximum within the subsequent 1 ps, indicating the completion of charge separation. During the evolution from LES to CSS, the hole absorption peak underwent spectral narrowing (Supplementary Fig. 27). This reflects the gradual enhancement of the cation feature, representing the evolution from partial charge transfer (CT) to complete charge separation[31,32]. Combined with the results of the kinetic analysis (Supplementary Fig. 28), the time required for this narrowing process is consistent with the time taken for hole absorption to reach its maximum. Hence, the fast charge separation is derived from partial CT state (CTS) dissociation in the film. Importantly, the consistent kinetics of the cation and anion, as well as their incomplete deactivation within

5 ns, indicates the presence of long-lived CSS (Fig. 3h and Supplementary Fig. 29). Nanosecond transient absorption (ns-TA) further highlighted this aspect (Fig. 3i–l), characterized by the persistent IDTT$^{\cdot+}$ and $C_{60}^{\cdot-}$ absorption, along with the GSB within 200 µs. The decay kinetics of radical ions at 200 µs timescale obtained from ns-TA provided a more intuitive identification of the long-lived CSS (Fig. 3l), which was absent in solutions (Supplementary Figs. 30 and 31). We fit the TA spectra for all the probe wavelengths available and obtained the decay-associated difference spectra (DADS) and corresponding time-constants (Fig. 3 and Supplementary Figs. 32–35): $\tau_{CS} = 0.4$ ps and $\tau_{CR} = 48.9$ µs for ID, and $\tau_{CS} = 0.7$ ps and $\tau_{CR} = 25.8$ µs for IB. The fitted nanosecond TAS provide a clearer view of the long-lived absorption features of IDTT$^{\cdot+}$ and $C_{60}^{\cdot-}$ at CSS (Supplementary Fig. 36)[3,12]. These results demonstrate that aggregation promotes concomitant fast CS and long-lived CSS.

## Fast charge separation promoted by hot CTS in aggregation

Fast CS in film is closely related to aggregation. In PMMA-diluted films, the CS rate significantly decreases, and the kinetics of excited-state resemble those of individual molecules in toluene (Fig. 4a and Supplementary Fig. 37), indicating that aggregation accelerates the CS process. We note that CSS in solution evolves slowly from lower-energetic LES. While the CS rate is remarkably faster in film, which possibly originates from higher-energetic states, such as the hot charge-transfer state (hot CTS) that has been confirmed by the spectral narrowing process[16,19,33]. Employing transient absorption under various excitation wavelengths (410, 520, 570 and 650 nm), it can be found that CS is gradually accelerated as the excitation wavelength blueshifts (Fig. 4b and c and Supplementary Fig. 38). For instance, the CS rate of 410 nm excitation is 100 times faster than that of 650 nm excitation. To

examine the intrinsic mechanism of CS acceleration, we analyzed the spectral evolution process within the first tens of picoseconds after excitation. Higher-energy photons (e.g. 410 nm) can excite electrons into more energetic LES, and then rapidly evolves into a hot CTS within 0.1 ps. The latter undergoes charge separation within 1 ps. In contrast, lower-energy photons hardly excite the electrons into sufficiently hot state; instead, the TA spectrum within 0.1 ps already exhibits partial hole absorption features (Fig. 4d, Supplementary Figs. 39, 40). Subsequently, its intensity gradually increases, accompanied by spectral narrowing (Fig. 4e–g, Supplementary Fig. 41), reflecting the evolution from CTS to CSS[31,32]. Therefore, in aggregated films the CSS evolves from CTS, and only hot CTS obtained by high-energy photons excitation can exhibit the fast CS feature. This can be attributed to the enhanced delocalization of the hot CTS, along with the increased density of electronic-state after aggregation, which can reduce the Coulomb potential for charge separation and opening up more resonant pathways between CTS and CSS[16,19,34,35]. For comparison, due to the lack of aggregation in solution, the CS rate is independent of the excitation wavelength (Supplementary Fig. 42). Hence, CS acceleration can be promoted by the hot CTS after aggregation.

## Long-lived intramolecular CSS mediated by free carriers

Now we turn to the question of what facilitates the long-lived CSS in the dyad aggregates. The "hot CTS" has been confirmed to accelerate CS in aggregates, which holds promise for dissociating into intermolecular free carriers (FCs). This contributes to the long-lived CSS, owing to the spin-uncorrelated FCs recombination[12]. To examine this point, we study the dependence of CR kinetics on excitation density (Fig. 5 and Supplementary Fig. 43). Charge recombination is accelerated with the increasing excitation density within 5 ns, indicating that

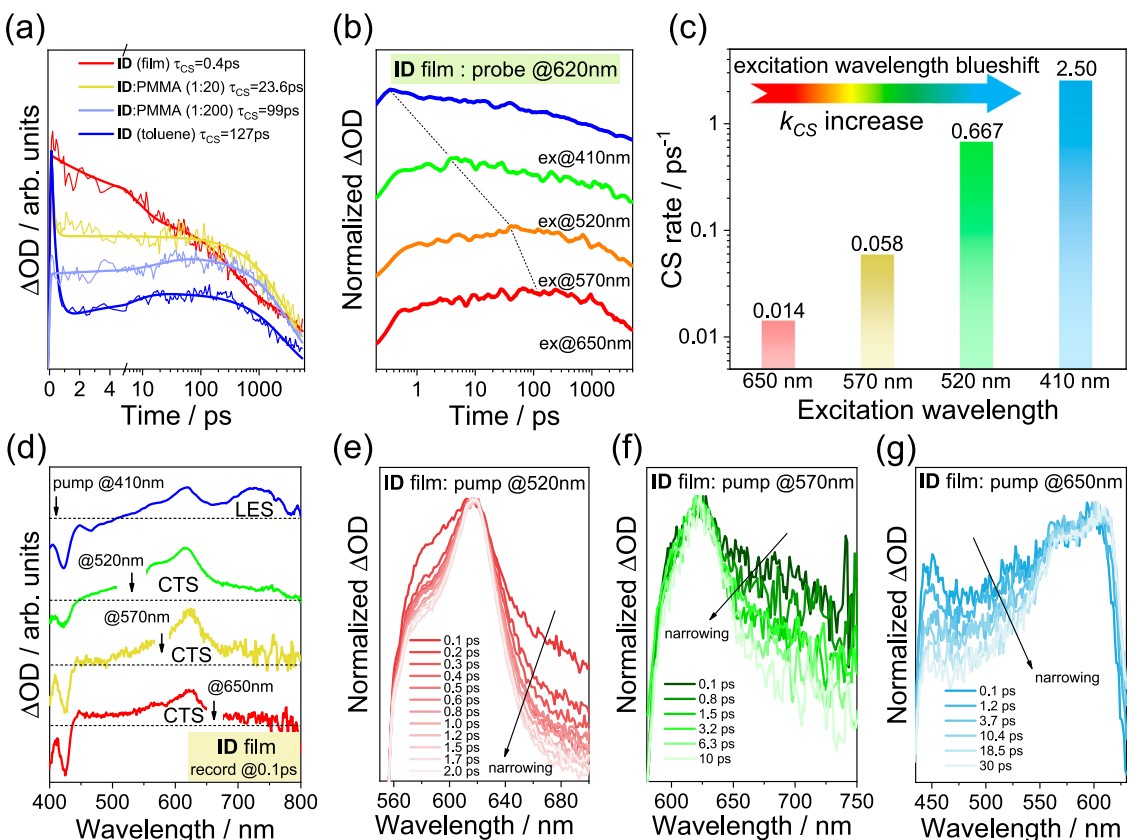

**Fig. 4 | Transient absorption and kinetics analysis under different wavelengths excitation. a** Charge-separated state (CSS) kinetics of ID film, PMMA-diluted ID film, and ID in toluene under 410 nm excitation. **b** Dependence of charge separation (CS) kinetics on the excitation wavelength. **c** CS rates under different wavelengths excitation. **d** Transient absorption spectrum within 0.1 ps when the ID film is excited at 410, 520, 570 and 650 nm (scattering background has been subtracted). **e–g** Spectral narrowing process of hole absorption peak when the ID film is excited at 520, 570 and 650 nm.

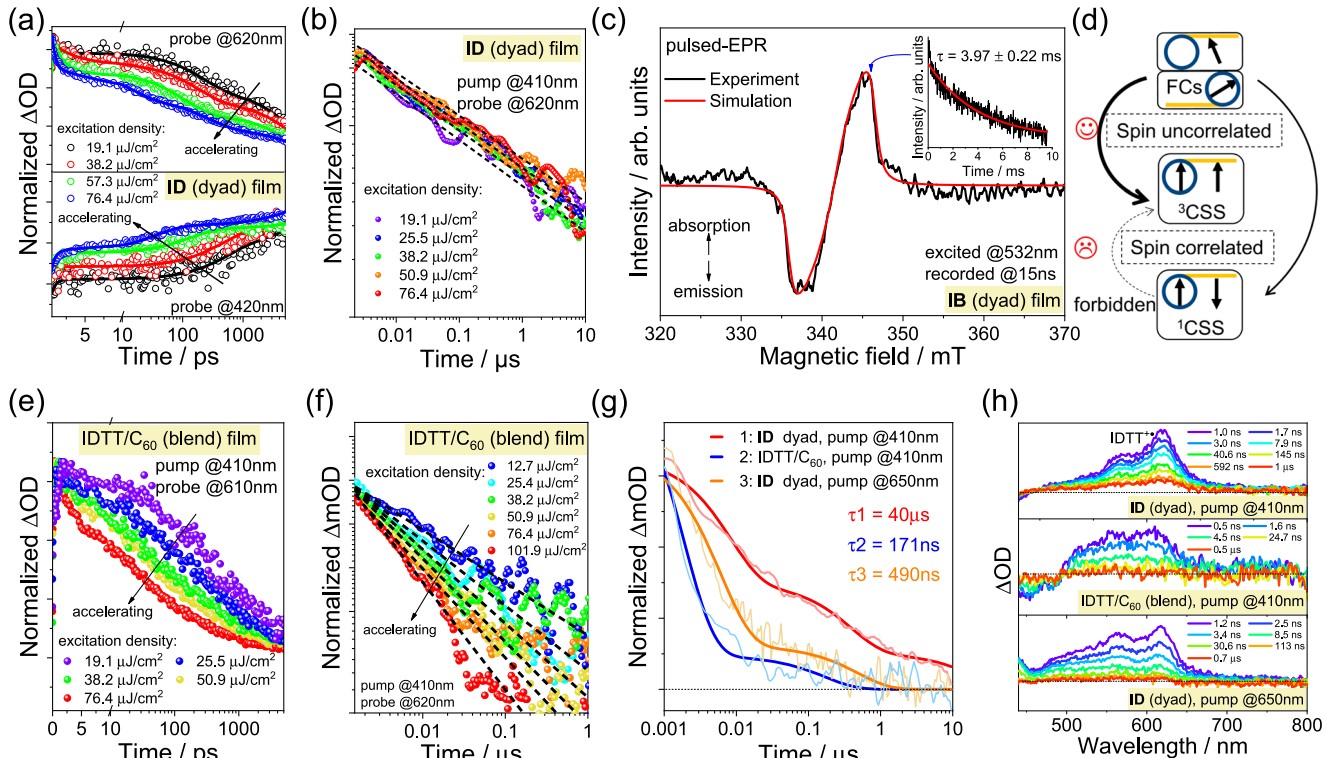

**Fig. 5 | TA kinetics and pulsed-EPR analysis. a** Dependence of the charge-separated state (CSS, probe at 620 nm) and ground-state bleaching (GSB, probe at 420 nm) kinetics on excitation density for ID dyad film within 5 ns. **b** Dependence of the CSS kinetics on excitation density for ID dyad film within 10 μs. The slope of the dashed line reflects the CR rate change. **c** Experimental and simulated pulsed-EPR signal recorded 15 ns after 532 nm excitation of IB aggregate at 5 K; inset: experimental and fitted decay kinetics. **d** Schematic of the generation of triplet charge-separated state ($^3$CSS) via spin-correlated $^1$CSS or spin-uncorrelated free carriers (FCs). **e, f** Dependence of the CSS kinetics on excitation density for IDTT/C$_{60}$ blend film within 5 ns and 1 μs. The slope of the dashed line in **f** reflects the CR rate change. **g** Comparison of the CSS kinetics between ID dyad and IDTT/C$_{60}$ blend films (19.1 μJ/cm$^2$ excitation) under 410 or 650 nm excitation. **h** Comparison of the selected TAS between ID dyad and IDTT/C$_{60}$ blend film within microsecond timescale.

CR is a bimolecular process[12,13]. That is, intermolecular CS exists in the films, consistent with the different luminescence and CS behaviors of the film before and after dilution (Figs. 2c and 4a). (i) The electron hole can entirely escape the Coulombic potential, resulting in complete separation into free carriers[21]. (ii) The intermolecular charge recombination process depends on the FCs density, which increases with the excitation density, resulting in accelerated CR rates[36]. When the timescale extends to the microsecond timescale, however, this dependence on the excitation density disappears, reflecting the intramolecular process. Persistent hole absorption and GSB, along with the continuous excited-state evolution prove that a considerable portion of the FCs recombine into intramolecular CSS. Such a long-lived CSS can be attributed to the formation of $^3$CSS via FCs mediation[8,37]. The spin-uncorrelated FCs in aggregates can readily recombine to form intramolecular $^3$CSS with a considerable proportion governed by spin statistics (Fig. 5d)[12,22]. For comparison, this dependence does not exist in solution or PMMA-diluted films (Supplementary Figs. 44–46), reflecting the lacking of FCs and resulting in the short-lived CSS lifetime. Furthermore, when employing low-energy photon excitation (e.g. 650 nm), the dependence of CR on excitation density decreases dramatically. This reflects the inhibition of the bimolecular recombination due to lower-populated CTS, aligning with the discussed "hot CTS". At this point, the generation of intermolecular FCs becomes inefficient, resulting in a significant decrease of the final CSS lifetime (490 ns for ID and 260 ns for IB under 650 nm excitation, Fig. 5g, Supplementary Figs. 47 and 48).

To further prove the final long-lived state is an intramolecular triplet CSS ($^3$CSS), we employ pulsed electron paramagnetic resonance spectroscopy (pulsed-EPR, Fig. 5c and Supplementary Fig. 52), which is a very sensitive approach to acquire the transient spin signal[38]. Using IB as an example, experiment and simulation indicate that a triplet state exists in the aggregate under 532 nm excitation, with the transient spin quantum number being $S = 1$. The absence of any $S = 1/2$ spin component indicates that no detectable separated intermolecular ion-pair exists in film after 15 ns. We then performed spin-Hamiltonian simulations on the triplet signal following Eq. 1[39,40]:

$$\widehat{\mathbf{H}} = \mu \mathbf{B} \mathbf{g}^= \widehat{\mathbf{S}} + \widehat{\mathbf{S}}^T \mathbf{D}^= \widehat{\mathbf{S}} \qquad (1)$$

where μ is the Bohr magneton constant, **B** is the experimental magnetic field, $\mathbf{g}^=$ is the Landé factor, with simulated principle values of 1.999710, 1.99985, 2.00243, $\widehat{\mathbf{S}}$ is the spin operator, and $\mathbf{D}^=$ is the zero-field splitting (ZFS) tensor, with $D$ and $E$ equal to −150 MHz and 50 MHz, respectively. These demonstrate that the final triplet is an intramolecular $^3$CSS rather $^3$C$_{60}$*, because of the relatively weaker spin-spin dipolar coupling revealed by the smaller ZFS parameters[41]. This is consistent with the ns-TA results (Supplementary Fig. 31). Additionally, we noticed that the signal intensity of the $^3$CSS is very weak, confirming that the $^3$CSS in aggregates are distinct from the spin-polarized signals obtained from intersystem crossing (ISC). And it is possibly due to the $^3$CSS generation mediated by spin-uncorrelated FCs recombination, resulting in lower spin polarization and further supporting the kinetic analyses obtained from transient absorption.

We note that the GSB decays following the FCs recombination, exhibiting excitation density dependence within 5 ns (Fig. 5a and Supplementary Fig. 43). This suggests the luminescent recombination of some FCs directly to the ground-state, which is consistent with the

intermolecular emission in the NIR region of the film, and the luminescence lifetime is matched with the measured intermolecular FCs lifetime (~ 5 ns). However, this is much shorter than the ultimate CSS lifetime. Therefore, this further demonstrates that, (i) an intermolecular CSS lasts about 5 ns in aggregates, and (ii) the final long-lived CSS involves microsecond-scale intramolecular CR and should belong to $^3$CSS[37]. In summary, these findings prove that aggregation facilitates the long-lived $^3$CSS generation mediated by the spin-uncorrelated FCs recombination.

### Comparison between dyad and blend aggregates

The special CS processes in D-A dyad aggregates inspires us to compare it with D/A directly blended systems. The steady-state spectra of the blend films are shown in Supplementary Fig. 10. In blends, charge recombination on both nanosecond and microsecond timescale is a bimolecular process, manifested by the CS rate acceleration with the increasing excitation density, likely due to the intermolecular electrons/holes diffusion (Fig. 5e, f and Supplementary Fig. 49)[42]. This depends more on the molecular order[3,43–45]. However, small molecules like IDTT, due to large steric hindrance of the side-chain and lack of π-extended end groups[46,47], are difficult to form ordered structures after blending with $C_{60}$, hence resulting in a final CSS lifetime of only 171 ns in IDTT/$C_{60}$ blends and 85 ns in IBDT/$C_{60}$ blends (410 nm and 19.1 μJ/cm$^2$ excitation, Fig. 5g and Supplementary Fig. 50). In contrast, the long-lived CSS in dyads is an intramolecular state. This significantly reduces the requirements for molecular order, making it easier to obtain long-lived CSS (40 μs) compared to blends[43]. We further prepared aggregated films of dyads using spin-coating, drop-casting and knife-coating, all exhibiting fast CS and long-lived CSS (Supplementary Fig. 51). Their similar spectra and kinetic evolutions further validate this phenomenon, which also differs from some reported fullerene/polymer blends[45,48]. From this perspective, dyad aggregates are more prone to achieving long-lived CSS compared to blends.

### Computational and photophysical processes analysis

Using DFT and time-dependent DFT, we analyzed the excited-state properties of the optimized monomers and dimers of ID and IB. The first 20 singlet excited-states (Fig. 6a and Supplementary Figs. 53–58) are assigned to LES and CTS according to the electron/hole distribution (Supplementary Note 10). For both dyads, the density-of-state for the dimers, especially CTS, is higher than the monomers, which can provide more resonant channels for accelerating CS. Then we compared the time taken for photoinduced CS and CR in several reported D-A systems[9,49], which highlights the prominent effect of the aggregation on CS regulation in IT-$C_{60}$ SMs (Fig. 6b). The concomitant fast CS and long-lived CSS in the dyad aggregates are more impressive compared to those of solutions, whose photophysical processes are schematically illustrated in Fig. 6c, d[12,13].

### Photoelectronic switch property

Based on the photophysical processes of the dyad aggregates, it can be observed that blue-light-excitation (e.g. 410 nm) can open up inter-molecular CS pathways, leading to long-lived intramolecular CSS and achieving fast CS meanwhile. Alternatively, this pathway can be closed using red-light-excitation (e.g. 650 nm) or in non-aggregated systems (e.g. solutions). This results in a two-order-of-magnitude decrease in the CS rate, and the ultimate CSS lifetime is shortened from 40 μs to 490 ns (650 nm excitation) and 311.4 ps (in solution, Fig. 5g and h). These properties render dyads having the potential of photoelectronic switches (Fig. 6e and Supplementary Fig. 59)[50–52]. Inputting different excitation wavelengths or aggregates, the pathway connecting inter-molecular to intramolecular CSS can be selectively opened or closed, further drastically regulating the CS rate and lifetime. This is one of the special features of the dyad aggregates, which can hardly be achieved in blends due to the lack of intramolecular CS processes.

### Photocatalysis applications

ID dyad aggregate has a longer CSS lifetime (40 μs) compared to blend (171 ns), giving dyad a better ability for redox equivalents storage confirmed by the photocurrent response[53]. We choose hydroquinone (HQ) as a photocatalytic substrate. As shown in Fig. 6, although the photocurrent density of the dyad and blend are similar when HQ is not added, upon the addition of 50 mM HQ, both exhibit enhancements in photocurrent. This indicates the additional charge exchange occurring between the electrode and the electrolyte containing HQ under illumination, resulting in the photocurrent increase via proton-coupled electron transfer (PCET) mechanism[53]. Importantly, dyad exhibits a six-fold enhancement, whereas blend shows only a 2.4-fold enhancement, indicating that dyad aggregates have a superior capability for redox equivalents storage. Similar results can be observed between IB dyad (Supplementary Fig. 60). This suggests that IT-$C_{60}$ dyad aggregates with long-lived CSS hold greater promise for the photocatalysis application compared to the blends.

## Discussion

In conclusion, aiming at efficient light-energy conversion, we revealed the role of aggregation in promoting photoinduced charge separation in organic small molecules. Aggregation enhances intermolecular interactions, forming the basis for hot CTS. High-energy photons excitation, opening up additional resonant channels between hot CTS and CSS, significantly accelerates the inter-molecular CS process. Using blue-light-excitation, the CS rate of the aggregates was approximately 100 times higher than that of the solution or red-light-excitation. Based on the dependence of CR kinetics on the excitation density, the long-lived CSS is an intramolecular state. But the intermolecular FCs mediation is necessary, where spin-uncorrelated electron/hole facilitate the intramolecular long-lived $^3$CSS generation demonstrated by pulsed-EPR. Thus, the CSS lifetime of the aggregates was approximately 10$^5$ times longer than solution. The special CS processes in dyad give it properties akin to a photoelectronic switch. Blue-light-excitation of dyad aggregates can open pathways from intermolecular to intramolecular CSS, enabling the simultaneous fast CS and long-lived CSS. This differs from aggregates directly blended donor and acceptor, wherein only intermolecular CS and CR occur. Therefore, the long-lived CSS necessitates higher demands for molecular order. In contrast, the intramolecular CSS in dyad significantly reduces these requirements, making it easier to obtain long-lived CSS. The prolonged CSS lifetime endows dyad aggregates with superior capabilities for redox equivalents storage, holding critical significance for photocatalysis. Therefore, this work explores the role of aggregation in promoting fast CS and long-lived CSS, reveals the unique CS mechanism in D-A dyad promoted by aggregation, and provides an important foundation for designing SM-based materials for the photocatalysis and photoelectronic devices applications.

## Methods
### Materials

The IT-CHO precursors were purchased from *Organtec Ltd*. The synthetic methodology for IT-$C_{60}$ according to the prato reaction is illustrated in the Supplementary Note 1. The thin-films were prepared by spin-coating from IT-$C_{60}$ solution in toluene (20 mg/mL) onto a quartz substrate at 3000 rpm. For the PMMA-diluted films, ID (IB) was mixed with PMMA according to a certain weight ratio (1:5, 1:20, 1:200 and 1:1000), and then heated and stirred in $o$-DCB at 60 °C for 4 h until completely dissolved and mixed. Finally, the ID (IB)/PMMA blend solutions were spin-coated onto quartz at 3000 rpm. For blend systems, an equal amount of $C_{60}$ and IDTT (IBDT) are mixed in $o$-DCB (20 mg/mL), and then the mixed solution is spin-coated on the quartz at 3000 rpm.

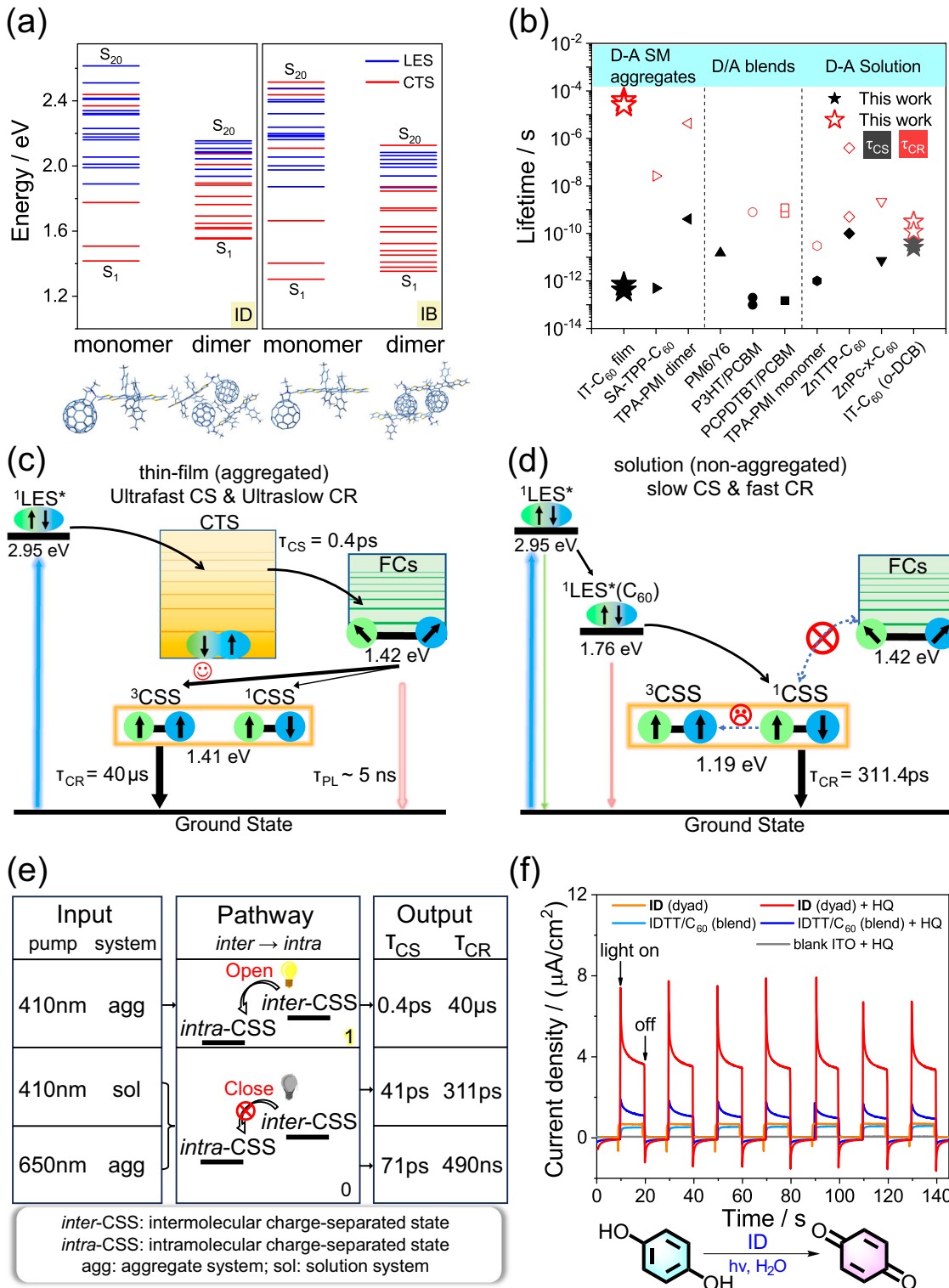

**Fig. 6 | Photophysical processes and application analysis. a** Singlet excited state distribution for ID and IB in monomer and dimer structures; red represents the first 20 singlet states with charge transfer characteristics (CTS), and blue represents the localized excited states (LES). **b** Comparison of the charge separation/recombination (CS/CR) kinetics among some representative photoelectronic materials. The black symbols represent the time taken for CS, red symbols represent the CSS lifetime, and different shaped symbols represent the photoelectronic materials listed on the X-axis. **c, d** Proposed schematic of excited states evolution for ID film and $o$-DCB under 410 nm excitation. Arrows and circles represent the spin orientation of singlet charge-separated state ($^1$CSS) or triplet charge-separated state ($^3$CSS) and charge separation degree respectively, wherein the free carriers (FCs) are spin-uncorrelated. **e** Schematic diagram of photoelectronic switch in ID dyad. **f** Photocurrent response for ID dyad and IDTT/$C_{60}$ blend films in 0.2 M $Na_2SO_4$ containing 50 mM hydroquinone (HQ) under AM 1.5 illumination (100 mW) at 0 V vs. Ag/AgCl; the structures of HQ and benzoquinone are shown below.

## Purification and characterization

The crude IT-$C_{60}$ products were separated using a silica column (eluent: petroleum ether and toluene 3:1) and further purified by high-performance liquid chromatography (HPLC) using a Buckprep column (10 mm × 250 mm) with toluene as the eluent at a flow rate of 6 mL/min. $^1$H NMR was carried out in $CDCl_3$ on an Avance-400 (400 MHz) spectrometer (Bruker) at 298 K with chemical shifts (δ, ppm) reported relative to the solvent peak. Mass spectrometry was performed using a MALDI-FT-ICR-MS spectrometer (Solarix, Bruker).

**AFM**. The atomic force microscope (AFM) images were acquired by using an insulating silicon AFM tip (OPUS, $k = 26 \, N m^{-1}$, $f_0 = 300 \, kHz$) to scan the surface of the ID and IB films at a scan rate of 0.977 Hz in the tapping mode.

**TEM**. Transmission electron microscopy (TEM) and high-resolution transmission electron microscopy (HRTEM) were performed using JEOL-2100F and FEI Tecnai F20 at 200 kV.

## Steady-state spectra

The steady-state absorption spectra were recorded using a UH4150 spectrophotometer (HITACHI). Photoluminescence (PL) spectra in the NIR region were recorded on an FLS1000 steady-state transient fluorescence spectrophotometer (Edinburgh Instruments) equipped with an NIR PMT detector (R5509). The PL lifetimes were measured by FLS980 using a visible detector (R928) and time-correlated single photon counting (TSCPC). PL spectra in the visible region (400–900 nm) were recorded using a Horiba spectrofluorometer (Fluoromax).

## Electrochemical measurement

Differential pulse voltammetry (DPV) measurements were performed using a CHI760E electrochemical analyzer (Chenhua, Shanghai) in a three-electrode configuration at a scan rate of $0.1 \, V \, s^{-1}$. In solution, measurements were carried out in 0.05 M tetrabutylammonium hexafluorophosphate ($TBAPF_6$) in deaerated $o$-DCB. A glassy carbon electrode was used as the working electrode, a platinum wire as the counter electrode, and a saturated Ag/AgCl electrode as the reference electrode. For the films, ITO glass coated with the samples was used as the working electrode, Pt as the counter electrode, and Ag/AgCl as the reference electrode. Measurements were carried out in 0.05 M $TBAPF_6$ in deaerated acetonitrile to prevent the sample from dissolving. All results were corrected using a ferrocene/ferrocenium couple (Fc/Fc$^+$).

## Photocurrent measurement

The photocurrent measurement was conducted within a quartz electrolytic cell, with the sample coated on ITO glass serving as the working electrode, and then immersed into the electrolyte containing a 0.2 M aqueous solution of $Na_2SO_4$. Ag/AgCl was used as the reference electrode, and a platinum wire was used as the counter electrode. The selected light source was a xenon lamp supplemented with an AM 1.5 filter (100 mW). The hydroquinone (HQ, 0.05 M) was chosen as a photocatalytic substrate. The photocurrent was measured separately before and after adding HQ.

## Transient absorption measurement

Ultrafast transient absorption spectroscopy (Ultrafast System, Helios and EOS) was performed at the Institute of Physics and Chemistry, Chinese Academy of Sciences[4]. A femtosecond laser (Coherent Inc.) delivered 25 fs pulses at 1 kHz, and the output was split to generate a white-light continuum. The excitation wavelength was obtained using tunable optical parametric amplifiers (TOPAS-C; Light Conversion). The specific excitation wavelengths and densities are described in the main text. The continuum was used as a broadband optical probe from the near-UV to near-IR regions. The probe from 350 to 750 nm was generated by focusing the fundamental laser beam onto a 3 mm $CaF_2$ plate, which was oriented and continuously shifted in perpendicular directions. A near-infrared probe was generated by focusing the beam on the YAG crystal. The TA spectrum was calculated from consecutive pump-on and pump-off measurements and averaged over 400 shots. The nanosecond transient absorption was measured by the EOS detection system, wherein a photonic crystal super-continuous nanosecond laser was used. The photonic crystal was excited by the Nd:YAG laser to produce a 2-kHz broadband detection light with a detection spectrum that covers the range of 360–1750 nm and has a pulse width of less than 1 ns. The decay-associated difference spectra (DADS) and evolution-associated difference spectra (EADS) were obtained using GloTarAn, a program based on the R package TIMP and singular value decomposition[54,55].

All samples used for TA measurements had an absorbance of approximately 0.7 OD (in a quartz cuvette with 1 mm optical path) at their maximum absorption wavelength, and the solution sample concentrations were approximately $10^{-4}$ M. The steady-state absorption spectra were employed before and after each measurement to ensure that no remarkable photodegradation occurred during the TA measurement.

## Pulsed electron paramagnetic resonance spectroscopy

All pulsed EPR experiments were measured on X-band Chinainstru&Quantumtech (Hefei) EPR100 spectrometer, started with photo-excitation by a nanosecond laser pulse with 532 nm wavelength (CNIlaser company)[40]. The full width at half maximum was 7 ns and the energy is 200 mJ per pulse. The EPR signal of the excited spin were recorded by the Hahn-echo method while the microwave frequency is 9.56 GHz, and π/2, π pulses and τ interval time equal to 15, 30 and 3000 ns respectively. The field-dependent spectrum was acquired by sweeping the magnetic field. The lifetime data of the transient spin was acquired using a "Laser-delay time-π/2-τ-π-τ-echo" sequence, with the delay time between the photoexcitation laser pulse and the Hahn-echo detection being varied.

## Quantum chemical calculation

The monomer and dimer molecular structures were optimized using the Gaussian16 package at the B3LYP-D3BJ/6-31 G* level[56,57]. To simplify the calculation, the side chain $C_6H_{13}$ was replaced with $CH_3$. Consequently, the frontier orbitals were obtained. Taking the initial optimized monomer and dimer structures of IB and ID, time-dependent DFT was used to calculate the vertical excited states at the same level, and the Multiwfn program was employed to analyze the electron-hole distribution[58,59]. The charge transfer state (CTS) is the state that the electrons and holes are distributed separately among the acceptor and donor; whereas the localized excited state (LES) is the state with no obvious difference between electron and hole distribution. All of the atomic coordinates of the optimized structures can be found in Supplementary Data 1 (ID monomer), Supplementary Data 2 (ID dimer), Supplementary Data 3 (IB monomer) and Supplementary Data 4 (IB dimer).

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

## Acknowledgements

This work was supported by the National Natural Science Foundation of China (52322204, 52072374, received by B.W.), the National Key R&D Program of China (Grant No. 2022YFA1205900, received by C.-R.W.). B. Wu particularly thanks the Youth Innovation Promotion Association of CAS (Y2022015, received by B.W.). We sincerely appreciate Prof. Dirk M. Guldi at the Friedrich-Alexander-Universität Erlangen Nürnberg for his valuable suggestions. We sincerely thank Dr. Jing Li, Dr. Heng Lu and Dr. Mengdi Liu at the Technical Institute of Physics and Chemistry, Chinese Academy of Sciences (CAS) for the help of transient absorption measurements. We appreciate Dr. Jie Liu, Dr. Jun Wang, Dr. Xiang Wang and Chuang Wen in the Institute of Chemistry, CAS for the help of materials characterization. We would like to thank *Editage* (www.editage.cn) for English language editing.

## Author contributions

C.W., B.W. and C.-R.W. conceived the concepts. C.W., T.D., Y.J., J.T., Z.H. Y.S. and W.W. performed the synthesis and characterization of samples. C.W carried out the quantum chemical analysis. Y.L., S.Z., W.W. and R.W. provided valuable ideas. C.W., B.W., Y.L., S.Z. and C.-H.W. measured and analyzed the TA and pulsed-EPR results. B.W., Y.L., Y.C. and C.-R.W. polished the language and supervised this work. C.W. wrote this paper. All authors discussed the results and commented on the manuscript at all stages.

## Competing interests

The authors declare no competing interests.
