## [Peer Review File · Nature Communications]

Aggregation Promotes Charge Separation in Fullerene-Indacenodithiophene DyadReviewer #1 (Remarks to the Author):

This manuscript reports the charge separation and charge recombination of covalent fullerene (C60)-indacenodithiophene (IT) small molecules in solution and films. It was found the charge separation is faster in film, but the charge recombination is slower in film. In principal the results are interesting. Revision is required before consideration for acceptance.

The global fitting and target analysis should be presented in the main text of the main text of the manuscript, not in Supplementary Information;

The exceptionally long CS state lifetime may be due to the intermolecular ion pairs in the film, i.e. due to the diffusion of the cation or the anions, the ion pair may reside on different molecules. If this is the case, then it is within expectation that a long-lived CS state will be observed in the film sample. The authors should make in-depth analysis on this issue, necessary experiments should be done to clarify the intermolecular CS state or the intramolecular CS state. Transient absorption spectra can hardly discriminate these two cases.

Reviewer #2 (Remarks to the Author):

Wang et al submitted a manuscript entitled "The Pronounced Role of Aggregation in Promoting Charge Separation in Fullerene-Indacenodithiophene Film". They investigated photoinduced electron transfer in film containing fullerene (electron acceptor) and indacenodithiophene (electron donor) to give a long-lived charge separation. Photoinduced events were observed by time-resolved transient absorption spectroscopies very well, monitored the excited state and radical ion species. This paper is a nice piece for the publication in Nature Communications, however, some problems should be clarified by analyses of photoinduced dynamics. The key point is detection and monitoring of fullerene radical anion and donor radical cation at near-IR region. This paper is required to additional experiments such as transient absorption measurements at near-IR region before publication. Here are my comments as below.

(1) Authors discuss about the transient absorption change in visible region. Fullerene anion radical observed at 1000 nm should be detected and monitored. The evidence of formation of charge separation should be shown.

(2) Figure 6 for energy diagram, the exact energy values should be added in the scheme.

(3) The energy of charge separated state 1.85 eV electrochemically determined from the oxidation and reduction potential as shown in Figure 13 in SI. This value is higher than that of triplet excited energy of fullerene (1.5 eV). The final state may be triplet excited fullerene via charge recombination.

(4) The absorption band due to the triplet excited state of fullerene at 700 nm is observed at 1.2 ps as shown in Fig 3b. Then, the radical cation of IBDT is generated. The energy diagram may be wrong.

(5) In composite film, how about intermolecular behaviors? The charge separation may be intermolecular charge separation generated by strong aggregation interaction.

Reviewer #3 (Remarks to the Author):

In this paper, the authors demonstrate the effect that aggregation (or the formation of ordered films) of indacenedithiophene-C60 dyads has on i) accelerating photoinduced charge separation, facilitated by the formation of hot charge transfer states, ii) the extension of the lifetime of charge separated states thanks to the mediation of free carriers, whose recombination gives rise to a statistically driven large ratio of triplet charge separated states.

The manuscript is well-written, the conclusion extracted from the different photophysical studies and DFT calculations are sound, the methodology is described in detail and the analyses support the conclusions. The results are interesting because the very different charge separation and charge recombination kinetics provides long lived charge separation lifetimes. However, I believe that the results rely on processes already reported, namely, fast charge separation promoted by hot charge separated states and the formation of long-lived charge separated state through aggregation, so the novelty of the findings is not that high for publication in Nature Communications. I miss a comparison between the results obtained in a blend of C60 and indacenedithiophene film and those obtained in the dyad, to give an added value to the finding, and some perspectives of a possible application of these molecules in real light-to-current conversion systems.

I would recommend the authors to send the work to a more specialized journal.

POINT-BY-POINT RESPONSES OF THE REVIEWER COMMENTS

Sincerely thanks to the reviewers' insightful and valuable comments, which are essential for enhancing the quality of our manuscript. We have supplied additional experiments, analyses and discussions. All the changes have been highlighted using red fonts in the revised manuscript and supplementary information, and the point-by-point responses are as follows. We hope our responses and revisions will meet the satisfaction of the reviewers.

Reviewer #1 (Remarks to the Author):

This manuscript reports the charge separation and charge recombination of covalent fullerene (C₆₀)-indacenodithiophene (IT) small molecules in solution and films. It was found the charge separation is faster in film, but the charge recombination is slower in film. In principal the results are interesting. Revision is required before consideration for acceptance.

The global fitting and target analysis should be presented in the main text of the main text of the manuscript, not in Supplementary Information;

Response:

Thanks for your suggestion. We have presented the results of the global fitting and target analysis in the revised manuscript (**Figs. 3c, g and k**).

The exceptionally long CS state lifetime may be due to the intermolecular ion pairs in the film, i.e. due to the diffusion of the cation or the anions, the ion pair may reside on different molecules. If this is the case, then it is within expectation that a long-lived CS state will be observed in the film sample. The authors should make in-depth analysis on this issue, necessary experiments should be done to clarify the intermolecular CS state or the intramolecular CS state. Transient absorption spectra can hardly

discriminate these two cases.

Response:

We sincerely thank you for the constructive questions. In this research, based on experimental results, we conclude that the microsecond-level long-lived charge recombination (CR) of aggregated films is an intramolecular process, yet its generation requires mediation through the nanosecond-level intermolecular charge-separated state (CSs). This conclusion was derived from the correlation between CSs kinetics and excitation density. Within the nanosecond timescale, the decay of CSs (probed at hole absorption) is accelerated with the increased excitation density, reflecting bimolecular charge recombination behavior. Meanwhile, the long-lived decay process within the microsecond timescale is independent of excitation density, indicating that the long-lived charge recombination is an intramolecular process (*Nature* **500**, 435-439 (2013); *Nature* **597**, 666-671 (2021); *J. Am. Chem. Soc.* **142**, 12751-12759 (2020)). Typically, the intermolecular charge separation (CS) feature is the concentration-dependent photophysical processes. Here, we utilized poly(methyl methacrylate) (PMMA) to diluted film, restricting intermolecular aggregation, and supplemented some additional photophysical experiments. More importantly, taking **IB** dyad for an example, we supplemented pulsed electron paramagnetic resonance spectra measurement of the aggregates, demonstrating that the ultimate long-lived CSs are the intramolecular triplet CSs.

(1) Confirmation of the Intermolecular Charge Separation.

Steady-state absorption spectra. The absorption features of the PMMA-diluted film (1:200) resemble those of individual molecule absorption in solution (**Fig. R1a**). However, for the pure films (**ID:PMMA=1:0**), the absorption spectra exhibits broadening and tailing features in the visible region, indicating that aggregation enhances intermolecular interactions. This provides the possibility for intermolecular charge separation.

Steady-state luminescence spectra. The luminescence spectra also show the similar emission behavior of PMMA-diluted films to solution under 400 nm excitation (**Fig.**

R1b), with distinct local emission peaks for donor IDTT (~500 nm) and acceptor C₆₀ (~720 nm). However, pure **ID** aggregated film does not exhibit local emissions, but rather display a significant broad emission peak in the NIR region (700–1200nm, **Fig. R1c**). This indicates that the emission in the NIR region of the pure film belongs to an intermolecular process, due to the absence of this NIR emission after dilution. Different from the local emission characteristics of the donor or acceptor, this emission process should be attributed to the radiative recombination of intermolecular electrons and holes (free carriers).

Fig. R1 Steady-state spectra. **a** Absorption spectra of **ID** and the precursors IDTT and C₆₀ measured in *o*-DCB and the film, along with the PMMA-diluted **ID** film (ID:PMMA=1:200 (w:w)). **b** Photoluminescence spectra of **ID** in *o*-DCB, ID film and PMMA-diluted **ID** film (ID:PMMA=1:200 (w:w)), excited at 400 nm. **c** Photoluminescence spectra of **ID** in *o*-DCB and film in the near-IR region, recorded by an NIR detector.

Transient Absorption. Compared to pure aggregated films, the photophysical processes of PMMA-diluted films are similar to the individual molecules in toluene. Charge recombination is an intramolecular process within nanoseconds, manifesting as the lack of dependence between excitation density and the CR kinetics (**Fig. R2a**). Additionally, charge separation is significantly decelerated (**Fig. R2b**), and charge recombination is accelerated meanwhile in the diluted films (**Fig. R2c**). The films before and after dilution exhibit markedly distinct photophysical behaviors, suggesting the presence of intermolecular charge separation in pure aggregated films.

Fig. R2 Charge-separated states (CSs) kinetics. **a** Dependence of CSs kinetics on excitation density for **ID** film and PMMA-diluted **ID** film; pump at 410 nm and probe at 620 nm. **b** Kinetics of CSs in **ID** film, PMMA-diluted **ID** film, and **ID** in toluene within 5 ns timescale. **c** Kinetics of CSs in **ID** film and PMMA-diluted **ID** film, within 10 μs timescale.

(2) Demonstrating Long-Lived Charge Recombination as an Intramolecular Process.

Transient absorption. Despite the presence of intermolecular CS processes in the film, it was found that the ultimate long-lived CSs exhibited intramolecular charge recombination. If this microsecond-scale CSs originates from intermolecular ion pairs, where the electron and hole reside on different molecules, their recombination would accelerate with increasing excitation density, i.e., a bimolecular process. This phenomenon is foreseeable in the blend systems (*Nature* 500, 435-439 (2013); *Nature* 597, 666-671 (2021); *J. Am. Chem. Soc.* 142, 12751-12759 (2020)). Here we have demonstrated that, in the precursor donor (IDTT) and acceptor (C_{60}) blend film, the CR rate increases with the increased excitation density at the microsecond-scale (**Fig. R3b**). In contrast, for **ID** dyad films, the CR kinetic does not vary with excitation density in the microsecond timescale, indicating that the long-lived CSs in dyad films is an intramolecular charge recombination process (**Fig. R3a**).

Fig. R3 Charge-separated state and photoluminescence kinetics. **a** Dependence of CSs kinetics on excitation density for **ID** dyad films within 10 μs timescale; pump at 410 nm and probe at 620 nm. **b** Dependence of CSs kinetics on excitation density for IDTT/ C_{60} blend films within 1 μs timescale; pump at 410 nm and probe at 610 nm. **c** Photoluminescence kinetic of **ID** film excited at 405 nm and recorded at 820 nm.

Luminescence lifetime. Additionally, the detected lifetime of intermolecular charge carriers (~ 5 ns, **Fig. R3c**), obtained from luminescent kinetics, is consistent with the excitation-density-dependent CSs decay process, but is significantly shorter than the final CSs lifetime (40 μs). This indicates that the ultimate long-lived charge recombination process is the intramolecular behavior, and it should be attributed to the triplet CSs (*Angew. Chem., Int. Ed.* 135, e202216010 (2023)).

Pulsed electron paramagnetic resonance spectra. We further supplemented the pulse electron paramagnetic resonance spectra (pulsed-EPR) measurements of the aggregates, which can provide more evidence about long-lived intramolecular charge-separated state than transient absorption (**Fig. R4**). Using **IB** as an example, experiment and simulation indicate that a triplet state exists in aggregate under 532 nm excitation, with the transient spin quantum number $S=1$. The absence of any $S=1/2$ spin component indicates that no detectable separated intermolecular ion-pair exists in film after 15 ns delay. That is, the final long-lived CS state is an intramolecular triplet CS state, rather than due to the intermolecular ion pair diffusion. Based on the spin-Hamiltonian simulations, we further confirmed that the final long-lived state is the charge-separated state, according to the simulated weaker spin-spin dipolar coupling revealed by the smaller ZFS parameters. Additionally, the weak signal intensity of the ^3CSs confirms that the ^3CSs in aggregates are distinct from the spin-polarized signals obtained from

intersystem crossing (ISC), which is possibly due to the ^3CSs generation mediated by spin-uncorrelated FCs recombination. We provided a detailed discussion of this aspect in the revised manuscript (**Fig. 5c**). Therefore, the pulsed-EPR measurements further support that the long-lived CS state in the film is an intramolecular triplet CSs, which requires the intermolecular FCs mediation of its generation.

Fig. R4 Pulsed electron paramagnetic resonance (pulsed-EPR). **a** Experimental and simulated pulsed-EPR signal recorded 15 ns after 532 nm excitation of **IB** aggregate at 5 K. **b** Experimental and fitted decay kinetic of the pulsed-EPR recorded at 345 mT.

In summary, the ultimate long-lived state is an **intramolecular** ^3CSs . Its formation requires the intermolecular CSs (FCs) mediation, where spin-uncorrelated charge carriers recombine mandated by spin-statistics, generating the long-lived intramolecular ^3CSs . Thank you again for the constructive questions and suggestions.

Reviewer #2 (Remarks to the Author):

Wang et al submitted a manuscript entitled "The Pronounced Role of Aggregation in Promoting Charge Separation in Fullerene-Indacenodithiophene Film". They investigated photoinduced electron transfer in film containing fullerene (electron acceptor) and indacenodithiophene (electron donor) to give a long-lived charge separation. Photoinduced events were observed by time-resolved transient absorption spectroscopies very well, monitored the excited state and radical ion species. This paper is a nice piece for the publication in Nature Communications, however, some problems should be clarified by analyses of photoinduced dynamics. The key point is detection and monitoring of fullerene radical anion and donor radical cation at near-IR region. This paper is required to additional experiments such as transient absorption measurements at near-IR region before publication. Here are my comments as below.

Response:

Thank you for your affirmation of this work. We have supplemented the transient absorption measurements at near-IR region TA experiments, and attributed the excited-state absorption of fullerene radical anion and donor radical cation, further confirming the occurrence of charge separation. Additional experimental supplements will be discussed specifically in the following sections.

(1) Authors discuss about the transient absorption change in visible region. Fullerene anion radical observed at 1000 nm should be detected and monitored. The evidence of formation of charge separation should be shown.

Response:

We sincerely thank you for the valuable suggestions. **Fig. R5** depicts transient absorption spectra (TAS) in the visible and near-IR region regions, where the excited-state absorption (ESA) of the radical cations IBDT⁺ (610, 680, 1200 and 1450 nm), IDTT⁺ (560, 620 and 1250 nm), and the anion C₆₀⁻ (around 1020 nm, **Fig. R5b, d, f**,

h) are clearly visible. **Fig. R6** illustrates the ESA kinetics of radical ions, as well as the evolution kinetics of ground-state bleach (GSB), where the increase and decrease in absorption intensity reflect the generation and recombination processes of charge-separated states. Typical absorption features of cation and anion, and the consistent kinetics, act as the evidence of the formation of charge separation. We have added the TAS in the NIR region in the revised version (**Fig. 3** and **Supplementary Fig. 19**).

Fig. R5 Transient absorption spectra. **a, c** Selected TA spectra (TAS) of femtosecond TA (fs-TA) excited at 410 nm of **IB** and **ID** film in the visible and near-IR region. **b, d** Amplified excited-state absorption spectra of fullerene anion radical around 1020 nm of **IB** and **ID** film. **e, g** Selected TA spectra (TAS) of femtosecond TA (fs-TA) excited at 410 nm of **IB** and **ID** in *o*-DCB in the visible and near-IR region. **f, h** Amplified excited-state absorption spectra of fullerene anion radical around 1020 nm of **IB** and **ID** in *o*-DCB.

Fig. R6 Time-absorption profiles at specific wavelengths and fitted curves. **a** IB film. **b** ID film. **c** IB in *o*-DCB. **d** ID in *o*-DCB.

(2) Figure 6 for energy diagram, the exact energy values should be added in the scheme.

Response:

Thank you for pointing out this insightful suggestion. Based on the Rehm-Weller equation, and combined with the steady-state spectra, electrochemical measurements, and geometry optimization, we have estimated the specific energy levels of the excited states, and added them in the schematic of excited states evolution (**Fig. R7**). For example, the localized excited state (LEs) level of the donor can be estimated by the energy corresponding to the intersection point of the normalized absorption and emission spectra (*J. Am. Chem. Soc.* 141, 12789–12796 (2019)). And the FCs energy level, i.e., intermolecular CSs, can be estimated from the emission peak of the film in the NIR region (*J. Am. Chem. Soc.* 134, 19661–19668 (2012)). We have added some discussion in the **Supplementary Figs. 8, 12 and 13**, and **Supplementary Tables 1 and 2** in the revised Supplementary Information.

Fig. R7 Proposed schematic of excited states evolution. a Aggregated **ID** films. **b** Non-aggregated **ID** in *o*-DCB solutions under 410 nm excitation. **c** Aggregated **IB** films. **d** Non-aggregated **IB** in *o*-DCB solutions under 410 nm excitation.

(3) The energy of charge separated state 1.85 eV electrochemically determined from the oxidation and reduction potential as shown in Figure 13 in SI. This value is higher than that of triplet excited energy of fullerene (1.5 eV). The final state may be triplet excited fullerene via charge recombination.

Response:

Thank you for the valuable question. We have demonstrated that charge recombination ultimately leads to the ground state, rather than to the triplet excited fullerene (${}^3\text{C}_{60}^*$), following the analysis of the transient absorption spectra, photoluminescence, electrochemical analysis, as well as the pulsed electron

paramagnetic resonance (pulsed-EPR) measurements.

Transient absorption spectra. Firstly, according to the transient absorption spectra, the decay of excited states in aggregated film mainly manifests as the hole absorption deactivation, accompanied by the decay of ground-state bleach (GSB). During this process, there is no absorption feature around 700 nm corresponding to the $^3\text{C}_{60}^*$ (**Fig. R8**), indicating the direct charge recombination to the ground state.

Fig. R8 Excited-state evolution during charge recombination within 5 μs . a, b Selected transient absorption spectra (TAS) and the normalized TAS of **ID** film during charge recombination, and the TAS of the $^3\text{C}_{60}^*$ obtained in toluene. c, d Selected transient absorption spectra (TAS) and the normalized TAS of **IB** film during charge recombination, and the TAS of the $^3\text{C}_{60}^*$ obtained in toluene.

Photoluminescence analysis. Additionally, the photoluminescence spectra confirm emission peak around 900 nm in the film (**Fig. R9**), attributed to intermolecular charge recombination, which is lower in energy than the $^3\text{C}_{60}^*$ emission (~ 827 nm), suggesting that charge recombination to the $^3\text{C}_{60}^*$ is not thermodynamically feasible.

Fig. R9 Photoluminescence spectra in the NIR region. a ID film following 400 nm excitation. b IB film following 400 nm excitation.

Electrochemistry analysis. Furthermore, the energy levels of CSs are estimated based on the redox potentials of precursor molecules of the donor (IDTT or IBDT) and acceptor (C_{60}), rather than the covalent derivatives (*Nat. Chem.* **6**, 899-905 (2014); *Z. Phys. Chem.* **133**, 93-98 (1982)). In the revised Supplementary Information, we retested the electrochemical properties of the precursor donor and acceptor, along with the dyad in *o*-DCB and film, and further estimated the energy levels of the charge-separated states (**Supplementary Figs. 12 and 13**, and **Supplementary Tables 1 and 2**). The estimated results from electrochemical measurements and photoluminescence spectra indicate that the energy levels of CSs are all below 1.5 eV (**Fig. R7**). Thus, the final evolution process should be the direct charge recombination to ground state.

Supplementary Fig. 12 Differential pulse voltammograms (DPV) recorded in *o*-DCB containing 0.05 M TBAPF₆ of the precursors, donor IDTT, IBDT and acceptor C₆₀. The ferrocene/ferrocenium (Fc/Fc⁺) was used as the internal standard reference.

Supplementary Table 2. Summary of the electrochemical properties of **ID** and **IB**, as well as the Gibbs free energies (ΔG) during charge separation (CS) and charge recombination (CR).

Energy level / eV	LEs donor	LEs acceptor	CSs in o -DCB	CSs in PhCN	CSs film	FCs
ID	2.95	1.76	1.19	0.93	1.41	1.42
IB	2.78	1.76	1.11	0.86	1.32	1.36

Pulsed-EPR measurements. We further supplemented the pulsed electron paramagnetic resonance spectra (pulsed-EPR) measurements of the aggregates, using **IB** molecules as an example. As shown in **Fig. R10**, experimental and simulation results indicate that, following 532 nm excitation at 5 K, the excited state at the microsecond timescale is a triplet state, with the spin quantum number $S=1$. We then performed spin-Hamiltonian simulations on the triplet signal:

$$\hat{H} = \mu \vec{B} \hat{g} \hat{S} + \hat{S}^T \hat{D} \hat{S}$$

where μ is the Bohr magneton constant, \vec{B} is the experimental magnetic field, \hat{g} is the Landé factor, with simulated principle values of (1.999710, 1.99985, 2.00243), \hat{S} is the spin operator, and \hat{D} is the zero-field splitting (ZFS) tensor, with simulated values of ($D=-150$ MHz and $E=50$ MHz). These demonstrate that the final triplet is an intramolecular ³CSs, rather ³C₆₀* ($D=-342$ MHz and $E=21$ MHz), because of the weaker spin-spin dipolar coupling revealed by the smaller ZFS parameters compared to ³C₆₀*, and the decay lifetime of the pulsed-EPR signal (~3.97 ms) prominently longer

than that of the ${}^3\text{C}_{60}^*$ (~ 0.41 ms, *J. Am. Chem. Soc.* 113, 2774–2776 (1991)). This is consistent with the ns-TA results (**Fig. R8**). The pulsed-EPR results and analyses have been added in **Fig. 5c** in the revised manuscript.

Fig. R10 Pulsed electron paramagnetic resonance (pulsed-EPR). **a** Experimental and simulated pulsed-EPR signal recorded 15 ns after 532 nm excitation of **IB** aggregate at 5 K. **b** Experimental and fitted decay kinetic of the pulsed-EPR recorded at 345 mT.

Therefore, energy level estimation and experimental results demonstrate that, charge recombination to the ${}^3\text{C}_{60}^*$ is thermodynamically unfeasible, with the ultimate decay of the state being CSs recombination directly to the ground state. We have supplemented some discussion about these results in the revised manuscript.

(4) The absorption band due to the triplet excited state of fullerene at 700 nm is observed at 1.2 ps as shown in Fig 3b. Then, the radical cation of IBDT is generated. The energy diagram may be wrong.

Response:

Thanks for your insightful question. It is possible that the transient absorption spectrum (TAS) at 1.2 ps does not represent the fullerene triplet state (${}^3\text{C}_{60}^*$) absorption. On the one hand, the spectral characteristics of them are quite different; on the other

hand, the formation of the $^3\text{C}_{60}^*$ usually occurs on the nanosecond timescale due to spin-forbidden. Here, in the *o*-DCB solution of **IB**, we present the TAS at 1.2 ps when using 410 nm to excite the IBDT donor, and compare it with the TAS before 0.3 ps when the C_{60} is excited using 350 nm (**Fig. R11a**). The spectral shapes of them are consistent, exhibiting partial hole absorption features with peaks at 610 and 680 nm. However, this differs from the spectral features of $^3\text{C}_{60}^*$. When using low-polarity toluene as the solvent, the energy level of the charge-separated state is higher than that of the $^3\text{C}_{60}^*$, and the electron/hole eventually recombines to the $^3\text{C}_{60}^*$ (**Fig. R11**), with a wide absorption band near 700 nm, and generation begins only after approximately 5 ns. Therefore, the TAS at 1.2ps in the **IB** solution does not belong to the $^3\text{C}_{60}^*$. Similar results can be observed in the *o*-DCB solution of **ID** (**Figs. R11c and d**). When using 410 nm to excite IDTT donor, the TAS at 1.5 ps is almost identical to the TAS at 0.3 ps where C_{60} is excited using 350 nm (at the beginning of excitation). Whereas it differs from the TAS of the $^3\text{C}_{60}^*$, indicating that the TAS around the first 1 ps is not the $^3\text{C}_{60}^*$ feature.

In the **ID** (**IB**) solution, the TAS around 1.2 ps upon excitation of the donor, closely resembles the TAS at the beginning of the C_{60} excitation. Additionally, the luminescence spectra have confirmed the energy transfer process from the donor to the acceptor. Therefore, the TAS around the first 1 ps is likely the localized excited state (LEs) of C_{60} via energy transfer from the LEs of donor. The radical cations of donor are gradually generated thereafter, indicating that charge separation in the solution evolves from the singlet LEs rather than from the $^3\text{C}_{60}^*$. In the revised version, the TAS at 1.2 ps that you mentioned is included in **Fig. 3** and **Supplementary Fig. 19**.

Fig. R11 Comparison between the transient absorption spectra. a, c Comparison of the TAS at 1.2 ps upon 410 nm excitation and 0.3 ps upon 350 nm excitation for **ID** and **IB** in *o*-DCB, along with the TAS of ${}^3C_{60}^*$. **b, d** Selected TAS of the radical cation absorption evolution and the ${}^3C_{60}^*$ formation in the toluene solution of **ID** and **IB**.

(5) In composite film, how about intermolecular behaviors? The charge separation may be intermolecular charge separation generated by strong aggregation interaction.

Response:

Sincerely thank you for pointing out this insightful question. Experimental results indicate the presence of intermolecular charge separation (CS) in composite film, with a lifetime of approximately 5 ns. However, the ultimate long-lived charge recombination (CR) is an intramolecular process. In the revised manuscript, we have supplemented photophysical experiments on the diluted film and the blended system, as well as the pulsed electron paramagnetic resonance (pulsed-EPR) measurements for further elucidation.

1) Confirmation of Intermolecular Charge Separation.

Steady-state absorption and emission spectra. The absorption characteristics of the film after PMMA dilution resemble those of individual molecule absorption in solution, and the broad absorption band in the visible region decreased significantly, indicating that intermolecular interactions exist in the pure film, providing the possibility of intermolecular CS. (**Fig. R12a**). Furthermore, luminescence spectra reveal local emissions of donor IDTT and acceptor C₆₀ in PMMA-diluted films, with disappearance of emission peaks in the NIR region. Conversely, pure **ID** aggregated films exhibit no apparent local emission, but rather show significant broad emission peaks around 900 nm in the NIR region (700–1200 nm, **Figs. R12b** and **c**). Thus, emission in the NIR region of pure films is an intermolecular process. Differing from the local emission features, this emission should be attributed to radiative recombination of intermolecular separated electrons and holes.

Fig. R12 Steady-state spectra. **a** Absorption spectra of **ID** and the precursors IDTT and C₆₀ measured in *o*-DCB and the film, along with the PMMA-diluted **ID** film (ID:PMMA=1:200 (w:w)). **b** Photoluminescence spectra of **ID** in *o*-DCB, ID film and PMMA-diluted **ID** film (ID:PMMA=1:200 (w:w)), excited at 400 nm. **c** Photoluminescence spectra of **ID** in *o*-DCB and film in the near-IR region, recorded by an NIR detector.

Transient absorption measurements. The charge separation kinetics of PMMA-diluted films (ID:PMMA=1:200) no longer exhibit excitation density dependence (**Fig. R13a**), manifesting as intramolecular charge separation (monomolecular CS). At this point, the photophysical process resembles that of individual molecules in toluene, with

a decreased rate of charge separation (**Fig. R13b**), simultaneous shortening of CS state (CSs) lifetime (**Fig. R13c**). In contrast, in pure films, the rate of charge recombination on the nanosecond timescale accelerates with increasing excitation density, reflecting a bimolecular recombination process, indicating the presence of intermolecular CSs (*Nature* 500, 435-439 (2013); *Nature* 597, 666-671 (2021); *J. Am. Chem. Soc.* 142, 12751-12759 (2020)). Moreover, pure films exhibit fast CS (within 0.4 ps) and long-lived CSs (40 μ s) compared to the diluted films. Therefore, the distinct differences in photophysical processes observed in PMMA-diluted and undiluted films indicate the presence of intermolecular charge separation in the composite films.

Fig. R13 Charge-separated states (CSs) kinetics. (a) Dependence of CSs kinetics on excitation density for **ID** film and PMMA-diluted **ID** film; pump at 410 nm and probe at 620 nm. (b) Kinetics of CSs in **ID** film, PMMA-diluted **ID** film, and **ID** in toluene within 5 ns timescale. (c) Kinetics of CSs in **ID** film and PMMA-diluted **ID** film, within 10 μ s timescale.

2) Revealing the Long-lived Intramolecular Charge Recombination.

Charge recombination kinetics. Despite the presence of intermolecular charge separation in the aggregated film, the ultimate long-lived CSs was found to manifest as an intramolecular CR process. Specifically, the charge recombination in dyad films at the microsecond timescale does not exhibit excitation density dependence, indicating that long-lived charge recombination is an intramolecular process (**Fig. R14a**). In comparison, precursor blend films (equal amount of IDTT and C₆₀ mixture) exhibit an

accelerated charge recombination rate at the microsecond timescale with increasing excitation density, suggesting a bimolecular process, namely intermolecular charge recombination (**Fig. R14b**).

Fig. R14 Charge-separated states and photoluminescence kinetics. (a) Dependence of CSs kinetics on excitation density for **ID** dyad films within 10 μs ; pump at 410 nm and probe at 620 nm. (b) Dependence of CSs kinetics on excitation density for IDTT/C₆₀ blend films within 1 μs timescale; pump at 410 nm and probe at 610 nm. (c) Photoluminescence kinetic of **ID** film excited at 405 nm and recorded at 820 nm.

Luminescence and charge-separated state lifetime. Additionally, the detected luminescence lifetime of intermolecular charge carriers (~ 5 ns, **Fig. R14c**) is nearly identical to the time taken for the excitation-density-dependent charge recombination. But it is significantly shorter than the ultimate charge recombination lifetime (40 μs), indicating that the final long-lived charge recombination process is not an intermolecular behavior, but rather a decay process of the intramolecular triplet CSs (*Angew. Chem., Int. Ed.* 135, e202216010 (2023)).

Pulsed electron paramagnetic resonance spectra. The long-lived intramolecular charge recombination process can also be confirmed by pulsed-EPR. As shown in **Fig. R10**, experimental and simulation results indicate that, following 532 nm excitation at 5 K, the excited state after 15 ns of excitation is a triplet state, with the spin quantum number $S=1$. The absence of $S=1/2$ spin component indicates that no detectable intermolecular ion-pair exists in film after 15 ns. That is, the final long-lived state is an intramolecular triplet state. In addition, according to the spin Hamiltonian simulations on the triplet state signal abovementioned, we have proved that the final excited state

is the charge-separated state (CSs), rather than ${}^3\text{C}_{60}^*$. Therefore, the ultimate long-lived excited state in aggregates is an intramolecular triplet charged-separated state (${}^3\text{CSs}$). Furthermore, the weak signal intensity of the ${}^3\text{CSs}$ confirms that the ${}^3\text{CSs}$ in aggregates are distinct from the spin-polarized signals obtained from intersystem crossing (ISC). This is possibly due to the ${}^3\text{CSs}$ generation mediated by spin-uncorrelated FCs recombination, resulting in lower spin polarization and further supporting the kinetic analyses obtained from transient absorption.

In conclusion, dyad aggregated films exhibit intermolecular CSs on the nanosecond timescale, and intramolecular CSs on the microsecond time scale. The ultimate long-lived charge recombination is an intramolecular process, with the generation necessitating intermolecular CSs (free carriers) mediation. We have provided further explanations in the revised manuscript.

Reviewer #3 (Remarks to the Author):

In this paper, the authors demonstrate the effect that aggregation (or the formation of ordered films) of indacenedithiophene-C60 dyads has on i) accelerating photoinduced charge separation, facilitated by the formation of hot charge transfer states, ii) the extension of the lifetime of charge separated states thanks to the mediation of free carriers, whose recombination gives rise to a statistically driven large ratio of triplet charge separated states.

The manuscript is well-written, the conclusion extracted from the different photophysical studies and DFT calculations are sound, the methodology is described in detail and the analyses support the conclusions. The results are interesting because the very different charge separation and charge recombination kinetics provides long lived charge separation lifetimes. However, I believe that the results rely on processes already reported, namely, fast charge separation promoted by hot charge separated states and the formation of long-lived charge separated state through aggregation, so the novelty of the findings is not that high for publication in Nature Communications. I miss a comparison between the results obtained in a blend of C60 and indacenedithiophene film and those obtained in the dyad, to give an added value to the finding, and some perspectives of a possible application of these molecules in real light-to-current conversion systems.

I would recommend the authors to send the work to a more specialized journal.

Response:

Thank you very much for pointing out these critical issues. In the revised manuscript, we further analyzed the charge separation/recombination (CS/CR) mechanism and applications of the donor-acceptor (D-A) dyads, as well as their differences from blend aggregates. The novelty of this study lies in revealing an exceptionally unique process of photoinduced charge separation within organic D-A small molecule aggregates. Specifically, aggregation simultaneously promotes the ultrafast CS and long-lived CSs, and the long-lived CSs of D-A dyad aggregates is an intramolecular triplet charge-

separated state (^3CSs), which requires the intermolecular CSs mediation of its generation. This endows dyad aggregates with the photoelectronic switch property responsive to excitation wavelength. And it makes the dyad aggregates easier to obtain long-lived CSs compared to blends, due to reducing the dependence on molecular arrangement and order, thus enhancing the ability of redox equivalents storage for the photocatalysis applications. Here we will discuss these issues in more detail.

(1) Charge Separation Mechanism in **D-A Dyad Aggregates**.

Firstly, the mechanism of the ultrafast CS and long-lived CSs induced by dyad aggregation, may differ from those reported previously. We found that with the aggregation being inhibited by poly(methyl methacrylate) (PMMA) dilution, the CS rate significantly slows down (ranging from sub-picoseconds to tens of picoseconds), indicating that intermolecular interactions after aggregation are the key factor leading to ultrafast CS. Meanwhile, the more energetic excited-states (hot states), obtained by high-energy photon (e.g., 410 nm) excitation in the aggregates, also act as a crucial role in accelerating CS. As the excitation wavelength blueshifts from 650 nm to 410 nm, the time taken for charge separation decreases from 71.3 ps to 0.4 ps (**Fig. R15a**).

More importantly, we confirmed that the ultimate long-lived CSs on the microsecond timescale is an **intramolecular charge recombination (CR) process**, rather than intermolecular recombination (**Fig. R15b**). However, the generation of this long-lived CSs requires mediation by the nanosecond-lived intermolecular CSs (free carriers, FCs), where the spin-uncorrelated electron-hole recombination follows spin-statistic rules, leading to the considerable triplet CSs (^3CSs) generation, thus manifesting noticeably long-lived CSs in the aggregate system. In the revised manuscript, we supplemented the pulsed-EPR experiments, further demonstrating that the long-lived charge-separated states in dyad aggregates is the intramolecular ^3CSs , mediated by the spin-uncorrelated FCs recombination of its generation. In non-aggregated systems, such as solutions or PMMA dilutions, the lack of FCs mediation results in long-lived ^3CSs only accessible through spin-flip (ISC), which poses challenges in thermodynamic and kinetic competition due to the spin-forbidden transition. Thus, obtaining long-lived CSs is challenging (**Fig. R15c**). Therefore, utilizing aggregation to open up intermolecular

CS channels, it can facilitate the of long-lived ^3CSs generation mediated by the spin-uncorrelated FCs recombination.

Fig. R15 Charge separation/recombination kinetics analysis. (a) Dependence of charge separation kinetics on the excitation wavelengths in **ID** film. (b) Dependence of the charge-separated state kinetics on excitation density at nanosecond timescale (top) and microsecond timescale (bottom). (c) Comparison of the charge recombination kinetics in **ID** film and PMMA-diluted **ID** film within 10 μs .

(2) Comparison of the **D-A Dyad** and **Blend Aggregate** Systems.

Photophysical process comparison. We apologize for the omission of a comparison between the dyad and blend systems. Here we have realized the necessity of this comparison and have supplemented experiments on the blends (IDTT and C_{60} at a 1:1 ratio) to compare their properties. Firstly, in the blends charge recombination is a bimolecular process, manifested by the significant acceleration of the CS rate with increasing excitation density (**Fig. R16a**), likely due to the intermolecular diffusion of electrons and holes (*Proc. Natl. Acad. Sci. U. S. A.* 109, 13498-13502 (2012); *Science* 343, 512-516 (2014); *J. Am. Chem. Soc.* 136, 10024–10032 (2014); *Nat. Commun.* 5, 3119 (2014); *Chem. Rev.* 121, 8234–8284 (2021)). However, although the dyad aggregates exhibit bimolecular CR process within 5 ns (the CR rate increasing with the excitation density, **Fig. R15b**), on the microsecond timescale, the CR no longer shows excitation density dependence (**Fig. R16b**), indicating an intramolecular process (*Nature* **597**, 666-671 (2021); *Nature* **500**, 435-439 (2013)). This can also be confirmed

by the emission spectra of the dyad aggregates. The PMMA-diluted film lacks NIR emission, while the pure aggregated film shows obvious emission peak around 900 nm, significantly red-shifted when compared to the local emission peaks, suggesting that the emission in the dyad aggregates originates from intermolecular electron-hole recombination (**Fig. R16c**). However, this luminescence process has a lifetime about 5 ns (**Fig. R16d**), comparable to the time taken for intermolecular CR, but much shorter than the lifetime of the final CSs (40 μ s). This indicates that the final microsecond-level long-lived charge recombination in dyad is not an intermolecular process, but an intramolecular process.

Fig. R16 Comparison between the dyad and blend aggregates. **a** Dependence of the charge-separated state (CSs) kinetics on excitation density for IDTT/C₆₀ blend films within 1 μ s timescale; pump at 410 nm and probe at 610 nm. **b** Dependence of the charge-separated state kinetics on excitation density for **ID** dyad films within 10 μ s timescale; pump at 410 nm and probe at 620 nm. **c** Photoluminescence spectra of **ID** film, **ID** in *o*-DCB, and PMMA-diluted **ID** film, excited at 400 nm and recorded by visible detector. **d** Luminescent kinetic of **ID** film. **e** Comparison of the selected transient absorption spectra in **ID** dyad and IDTT/C₆₀ blend aggregates at microsecond timescale. **f** Comparison of the CSs kinetics between **ID** dyad and IDTT/C₆₀ blend

aggregates at microsecond timescale (19.1 $\mu\text{J}/\text{cm}^2$ excitation).

Advantages of dyad aggregates. It is noteworthy that the CSs lifetime in dyad aggregates (**40 μs**) remarkably surpasses that in blends (**171 ns**), possibly attributed to the different charge recombination mechanisms (**Figs. R16e and f**). In blends, long-lived charge diffusion/recombination tends to require ordered molecular stacking or arrangement to form ordered pathways (*Proc. Natl. Acad. Sci. U. S. A.* 109, 13498–13502 (2012); *Science* 343, 512–516 (2014); *J. Am. Chem. Soc.* 136, 10024–10032 (2014); *Nat. Commun.* 5, 3119 (2014); *Chem. Rev.* 121, 8234–8284 (2021)). However, in the small molecular systems we investigated, due to large steric hindrance of the side-chain and lack of π -extended end groups, forming ordered structures after blending with fullerenes are difficult (*J. Am. Chem. Soc.* 142, 652–664 (2020), *Adv. Electron. Mater.* 4, 1700410 (2018)). Thus, the ultimate CSs lifetime is limited at 171 ns. Differently, in dyad aggregates, despite the potential lack of ordered pathways, their long-lived CSs originates from the intramolecular electron-hole recombination, presenting a unique advantage compared to the blend systems. Specifically, the dyad aggregates significantly reduce the requirements for molecular arrangement and order, making it easier to obtain long-lived CSs compared to blends (*Chem. Rev.* 121, 8234–8284 (2021)). Therefore, the final CSs lifetime in the blend aggregates is only 171 ns, while it extends to 40 μs in the dyad aggregates. We have prepared aggregated films using spin-coating, drop-casting and knife-coating, all exhibiting features of both ultrafast CS and long-lived CSs (**Fig. R17**). Their similar spectra and kinetic evolutions further validate this phenomenon. Consequently, dyad aggregates are more prone to achieving long-lived CSs.

Fig. R17 Transient absorption spectra. **a-c** Selected TAS in visible region for **IB** film fabricated by spin-coating, drop-coating and knife-coating. **d** Normalized charge separation kinetics of **IB** film prepared by these different fabrication processes. **e-g** Selected TAS in visible region for **ID** film fabricated by spin-coating, drop-coating and knife-coating. **h** Normalized charge separation kinetics of **ID** film prepared by these different fabrication processes.

(3) Potential applications.

Photoelectronic switch. It has been observed that blue-light-excitation (such as 410 nm) of the dyad aggregates can open up the intermolecular charge separation pathway, leading to long-lived intramolecular charge-separated state (40 μ s), and achieving ultrafast charge separation (sub-picosecond) at the meanwhile. However, when using red-light-excitation (such as 650 nm), or in non-aggregated systems (solutions), this intermolecular CSs pathway can be limited, resulting in a two-order-of-magnitude decrease in the charge separation rate. And the ultimate CSs lifetime is shortened from 40 μ s to 490 ns (under 650 nm excitation) and 311.4 ps (in solution), as shown in **Fig. R18**. This property may be applicable in the fields such as molecular photoelectronic switches, by modulating the excitation wavelength or aggregation to open the additional pathways from intermolecular to intramolecular charge separation, thus achieving the desired ultrafast CS and long-lived CSs simultaneously. Similar results can be found in **IB** dyad. (*Nat. Chem.* (2024) doi:10.1038/s41557-023-01420-w; *J. Am. Chem. Soc.*

142, 12658-12668 (2020); *Nat. Commun.* 15, (2024) doi:10.1038/s41467-024-46646-5). A schematic diagram is provided in **Fig. R18c**. And these results have been added in the **Fig. 6e** and **Supplementary Fig. 57**.

Fig. R18 Properties of the photoelectronic switch in ID dyad. **a** CS rates under different excitation wavelengths, reflecting the accelerated charge separation with the excitation wavelengths blueshift. **b** Kinetics of charge-separated state within 10 μs timescale for ID dyad film under 410 nm and 650 nm excitation. **c** Schematic diagram of photoelectronic switch in ID dyad.

Transient redox equivalents storage for photocatalysis. The long-lived CSs in dyad aggregates has the better capability to the storage of redox equivalents, which is essential for efficient photocatalytic redox reactions. We employed the photocurrent responses measurements of dyad and blend films, with or without hydroquinone (HQ, a photocatalytic substrate), to evaluate their capacities for storing redox equivalents (*J. Am. Chem. Soc.* 140, 9823–9826 (2018)). As shown in **Fig. R19**, the photocurrent magnitudes of the two are similar when HQ is not added (approximately 0.6 μA/cm²

for **ID**). Upon the addition of 50 mM HQ, both exhibit enhancements in photocurrent density. However, **ID** dyad exhibits a 6-fold enhancement, whereas blend shows only a 2.4-fold enhancement, indicating that dyad aggregates with long-lived CSs have a superior capability for redox equivalents storage. Similar results can be obtained in **IB** dyad and blend aggregates. This suggests that D-A dyad aggregates with long-lived CSs can be utilized in the field of photocatalysis. We added these results in the **Fig. 6f** and **Supplementary Fig. 58** in the revised version.

Fig. R19 Photocurrent response. **a, b** Photocurrent response for ITO|**ID** and ITO|**IB** in 0.2 M Na₂SO₄ aqueous solution with or without 50 mM hydroquinone (HQ) under AM 1.5 illumination (100 mW) at 0 V vs. Ag/AgCl.

In summary, utilizing blue light to excite dyad aggregates can open up additional intermolecular CS pathways, enabling ultrafast CS and leading to long-lived ³CSs simultaneously. In this case, it would function as a molecular photoelectronic switch. Modulating the excitation wavelengths or aggregates, can open or close intra- or intermolecular charge separation pathways, as well as the corresponding time taken for charge separation/recombination. The mechanism of long-lived CSs in dyad aggregates is different from that of the blend system. It does not require charge diffusion in the ordered intermolecular pathways, but originates from the intramolecular long-lived charge recombination. This reduces the dependence of long-lived CSs on the morphology and preparation technology, making it easier for dyads to acquire long-lived CSs. Correspondingly, dyad aggregates with long-lived CSs are more conducive to storing redox equivalents, facilitating their use in photocatalytic reactions.

We would like to express our gratitude for your beneficial questions and suggestions once again, and sincerely hope that you are satisfied with our responses.

Reviewer #1 (Remarks to the Author):

The revision is satisfactory, I suggest acceptance of the manuscript in its current form.

One minor issue, I am afraid the authors did continuous wave (CW) time-resolved EPR spectra, and it is not pulsed EPR spectrum, these are two different techniques. Please clarify it if they mentioned it in the revision manuscript.

Reviewer #2 (Remarks to the Author):

Authors report a photoinduced long-lived charge separated (CS) state in a PMMA film containing fullerene-indacenodithiophene dyad. There is a serious problem which is the lifetime of the CS state of 40 microseconds as noted in abstract. The determination of CS lifetime should be carried out more carefully. This paper may be better to resubmit to other specialized journal in the field of photochemistry or physical chemistry.

Fig 3j: The transient absorption should be recorded with the NIR region. The C60 anion radical is appeared at 1000 nm.

The decay of IDTT cation radical in a film was monitored by 19.1 microseconds as shown in Fig 3j. The CS lifetime of 40 microseconds could not be determined from the shorter time range of than half-life. The value contains significant error. Authors should monitor CS state with longer time range (~100-200 microseconds).

Fig 3f: C60 anion radical is gone at 2 ns after laser excitation when IDTT cation radical is remaining.

Reviewer #3 (Remarks to the Author):

I am really impressed by the effort carried out by the authors to address all the aspects posed by the reviewers.

I accept the publication of the revised version

POINT-BY-POINT TO REVIEWER COMMENTS

We express our sincere respect and heartfelt gratitude to each reviewer for their professional comments. In the revised manuscript, we further clarified the pulsed-EPR testing method, and supplemented the nanosecond transient absorption spectra in the NIR region, while also extending the monitoring timescale to 200 μs in the transient absorption experiments. The experimental results indeed demonstrate the existence of long-lived C_{60} anions and IDTT cations in the aggregated films, with a fitted charge-separated state lifetime of approximately 40 μs (for **ID** films). The point-by-point responses are as follows. We hope our responses and revisions will meet the satisfaction of the reviewers.

Reviewer #1 (Remarks to the Author):

The revision is satisfactory, I suggest acceptance of the manuscript in its current form.

One minor issue, I am afraid the authors did continuous wave (CW) time-resolved EPR spectra, and it is not pulsed EPR spectrum, these are two different technique. Please clarify it if they mentioned it in the revision manuscript.

Response:

Thank you for pointing out the difference between CW TREPR and pulsed EPR. We have noticed that there are two distinct EPR techniques available to provide high time resolution with an EPR spectrometer, and we use pulsed EPR in this work. In practice, we performed both CW TREPR and pulsed EPR experiment to the sample to characterize the photoexcited triplet CSs. However, measuring the transient spin signal in a thin-film using CW TREPR was turned out to be insufficiently sensitive. This is understandable because the limited spin amount in the thin-film according to the kinetic procedure proposed in this work, and the magnetic modulation technique (crucial to gain a high-quality CW-EPR signal) was not applicable in the CW-TREPR mode. Therefore, we primary used a pulsed EPR to resolve the spin signal. The pulsed EPR

signal was acquired using a sequence of “Laser-wait- $\pi/2$ - τ - π - τ -Hahn Echo”. The pulsed EPR method is a very sensitive approach to acquire the transient spin signal in fullerene and its derivatives. It has been used in literatures (*Nat. Phys.* 8, 596–600 (2012); *J. Phys. Chem. B* 126, 10519–10527 (2022)) and in our previous works (*npj Quantum Inf.* 7, 32 (2021)). The pulsed EPR spectrum with better signal to noise ratio is more suitable to validate the discovery reported in this work, thus we used it in our manuscript.

Reviewer #2 (Remarks to the Author):

Authors report a photoinduced long-lived charge separated (CS) state in a PMMA film containing fullerene-indacenodithiophene dyad. There is a serious problem which is the lifetime of the CS state of 40 microsecond as noted in abstract. The determination of CS lifetime should be carried out more carefully. This paper may be better to resubmit to other specialized journal in the field of photochemistry or physical chemistry.

Response:

Thank you for your valuable questions. The lifetime of the CS state of 40 μ s mentioned in the abstract is detected in aggregated **ID** films, without PMMA doping. In the revised version, we retested the transient absorption spectra of the aggregated films, extending the monitoring timescale to 200 μ s and supplementing nanosecond transient absorption tests in the NIR region. The experimental results show the coexistence of absorption peaks of IDTT cations in the visible and near-infrared regions, as well as C₆₀ anions in the NIR region, with consistent kinetic processes. Based on the kinetic fitting, the lifetime of the final charge-separated state (CSs) is estimated to be approximately 40 μ s. We will provide specific analysis in the following discussion.

Fig 3j: The transient absorption should be recorded with the NIR region. The C₆₀ anion radical is appeared at 1000 nm.

Response:

Thank you for your insightful suggestions. In previous work, we measured the femtosecond transient absorption spectra, obtaining the absorption features of IDTT cations in the visible and near-infrared (NIR) regions (620 and 1250 nm), as well as C₆₀ anion in the NIR region (around 1020 nm). The absorption peak of C₆₀ anion is slightly red-shifted compared to the literature-reported value (1010 nm, *Acc. Chem. Res.* 33, 695–703 (2000); *J. Am. Chem. Soc.* 129, 51, 15865–15871 (2007)), which should be due to partial overlap with the absorption of IDTT cations. Based on the coexistence of cation and anion radicals, and the consistency of kinetics, it can be confirmed that charge separation occurs after photoexcitation. Subsequently, nanosecond transient absorption spectra (ns-TA) in the visible range of aggregated films were tested, confirming the existence of long-lived CSs based on the absorption peaks of cation radical and the kinetic evolution processes.

Herein, we further supplemented the ns-TA spectra of aggregated films in the NIR region. To be honest, due to our instrument and technical limitations, it is challenging to detect the nanosecond transient absorption spectra of samples in the NIR region. Thus, the quality of the measured spectral signals is not ideal, and the scattering light is difficult to subtract. Nevertheless, we can still observe the absorption features of C₆₀ anion and IDTT cation in the NIR region (Figs. R1a and R1b), and the fitted spectra more clearly exhibit the absorption features of cation and anion radicals, as shown in Fig. R1c (*Nature* 500, 435-439 (2013); *Science* 343, 512-516 (2014)). In addition, the ion radical kinetics detected in the NIR region exhibit an obvious long-lived decay process (Fig. R1d). Based on the spectra and kinetics in the visible and NIR regions, the decay processes of C₆₀ anion and IDTT cation are consistent within 200 μs, thus proving the existence of long-lived charge-separated state in the aggregated film. We have added the additional experimental results to Figs. 3i-l in the revised manuscript.

Fig. R1 (a) Contour plot of the nanosecond transient absorption (ns-TA) for **ID** film under 410 nm excitation. (b) Selected transient absorption spectra of **ID** film from 0.1 μs to 100 μs . (c) Experimental and fitted transient absorption spectra of **ID** film. (d) Kinetics comparison of the IDTT cation (probed at 620 nm and 1250 nm) and C_{60} anion (probed at 1020 nm), as well as the ground-state bleaching (probed at 435 nm) within the 200 μs timescale.

The decay of IDTT cation radical in a film was monitored by 19.1 microsecond as shown in Fig 3j. The CS lifetime of 40 microsecond could not be determined from the shorter time range of than half-life. The value contains significant error. Authors should monitor CS state with longer time range (~ 100 -200 microseconds).

Response:

Thank you for pointing out the key issues. Here we retested the nanosecond transient absorption spectra and extended the monitoring time to 200 μs . As you said, the

experimental results revealed the existence of long-lived CSs. Although weaker, the radical ions absorption can still be observed around 40 μs (Figs. R1a and R1b). In Fig. R1c, in addition to the original spectra, the fitted spectra are also shown, which more clearly reveal the absorption of cations at 620 and 1250 nm, and the absorption of C_{60} anion around 1000 nm (*Nature* 500, 435-439 (2013); *Science* 343, 512-516 (2014)). The more direct evidence of long-lived CSs is the decay kinetics within the 200 μs timescale (Fig. R1d). Based on the consistency of the decay processes of cation and anion radicals, combined with the fitting results, we confirm the existence of long-lived charge-separated state in the aggregated film, with a fitting result of approximately 48.9 μs . We have added the additional experimental results to Figs. 3i-l in the revised manuscript.

Fig 3f: C_{60} anion radical is gone at 2 ns after laser excitation when IDTT cation radical is remaining.

Response:

Sincerely thanks for your questions. Due to the relatively weak absorption coefficient of C_{60} anion (*Acc. Chem. Res.* 33, 695-703 (2000)), it may not appear very obvious in Fig. 3f in the previous manuscript. Here, we separately present the transient absorption spectra of the **ID** film from 0.3 ps to 2 ns, and magnify the absorption features of C_{60} anion around 1000-1100 nm. It can be seen that significant C_{60} anion ($\text{C}_{60}^{\bullet-}$) absorption features still exist after 2 ns. A more intuitive comparison can be observed from the kinetics. Fig. R2c shows the normalized decay kinetics of $\text{C}_{60}^{\bullet-}$ (probed at 1020 nm) and $\text{IDTT}^{\bullet+}$ (probed at 1250 and 620 nm), indicating consistent decay process of cation and anion radicals. This also indicates that IDTT cation and C_{60} anion coexist. Additionally, we compared the decay processes of $\text{C}_{60}^{\bullet-}$ in film and solution, as shown in Fig. R2d. The normalized decay kinetics show that $\text{C}_{60}^{\bullet-}$ in *o*-DCB solution almost decayed completely around 1 ns, while this decay process becomes faster in PhCN. However, $\text{C}_{60}^{\bullet-}$ is still not completely decayed within 5 ns in the aggregated film, retaining a significant amount of excited-state components. Based on the consistency

of the kinetics of $C_{60}^{\bullet-}$ and $IDTT^{+\bullet}$, and the presence of excited-state species not completely decayed within 5 ns in the **ID** film, it indicates the existence of long-lived charge-separated state. We further confirmed this long-lived CSs through nanosecond transient absorption techniques (Fig. R1).

In the revised version, we supplemented the kinetics of $C_{60}^{\bullet-}$ in Fig. 3d and Fig. 3h, and added the enlarged $C_{60}^{\bullet-}$ absorption spectra around 2 ns in Fig. 3f.

Fig. R2 (a) Selected transient absorption spectra of ID film under 410 nm excitation. (b) Amplification of C_{60} anion absorption from 900 to 1200 nm after 234 ps under 410 nm excitation. (c) Kinetics comparison between C_{60} anion ($C_{60}^{\bullet-}$, probed at 1020 nm) and IDTT cation ($IDTT^{+\bullet}$, probed at 620 and 1250 nm). (d) Kinetics comparison of the C_{60} anion in **ID** film, *o*-DCB and PhCN.

Therefore, combining the experimental results of the visible and NIR regions within 200 μ s, it can be proved that charge separation occurs in the aggregated film after

excitation, and IDTT cation and C₆₀ anion coexist, with a long-lived charge-separated state of about 40 μs. We would like to thank you again for your timely pointing out these key suggestions in our work, and hope that you can be satisfied with these results.

Reviewer #3 (Remarks to the Author):

I am really impressed by the effort carried out by the authors to address all the aspects posed by the reviewers.

I accept the publication of the revised version

Response:

We sincerely thank you for the valuable guidance you have given to this work, and we appreciate your positive comments here.

Reviewer #2 (Remarks to the Author):

Authors revised manuscript according to my comments. Fig 3 has been improved. The photophysical analyses is now acceptable. I think this paper may be publishable in the current form.